# PuzzleMoE: Efficient Compression of Large Mixture-of-Experts Models via Sparse Expert Merging and Bit-packed inference

## Abstract

Mixture-of-Experts (MoE) models have shown strong potential in scaling language models efficiently by activating only a small subset of experts per input. However, their widespread deployment remains limited due to the high memory overhead associated with storing all expert parameters, particularly as the number of experts increases. To address this challenge, prior works have explored expert dropping and merging strategies, yet they often suffer from performance drop at high compression ratios. In this paper, we introduce PuzzleMoE, a training-free MoE compression method that achieves both high accuracy and efficient inference through two key innovations: First, PuzzleMoE performs sparse expert merging by identifying element-wise weight redundancy and specialization. It uses a dual-mask to capture both shared and expert-specific parameters. Second, to avoid the overhead of storing binary masks and signs, PuzzleMoE introduces a bit-packed encoding scheme that reuses underutilized exponent bits, enabling efficient MoE inference on GPUs. Extensive experiments demonstrate that PuzzleMoE can compress MoE models by up to 50% while maintaining accuracy across various tasks. Specifically, it outperforms prior MoE compression methods by up to 16.7% on MMLU at 50% compression ratio, and achieves up to $1.28\times$ inference speedup.

## 1 Introduction

Mixture-of-Experts (MoE) architectures have demonstrated remarkable scalability in language models by activating only a subset of expert sub-networks per input. However, deploying these models in real-world applications remains challenging in practice due to their large memory footprint: all expert weights must be stored in memory, regardless of which subset is activated during inference. As the number of experts grows, as evidenced by recent advances such as Mixtral (Jiang et al., 2024), DeepSeek-MoE (Dai et al., 2024), and Qwen-MoE (Team, 2024), this memory overhead becomes increasingly prohibitive, especially for resource constrained deployments. For instance, Mixtral-8x7B has 47 billion parameters, about 45 billion of which are in the expert modules, requiring at least two A100-80GB GPUs to load in Bfloat16.

To reduce this memory cost, previous studies have investigated bit-level compression techniques, such as quantization, to eliminate redundancy within individual experts (Hu et al., 2025; Huang et al., 2025). However, these methods are designed to eliminate the redundancy of individual experts without considering the redundancy across different experts (He et al., 2025a). Recent works have thus focused on expert dropping and expert merging. Expert dropping methods (Lu et al., 2024; Lee et al., 2025b) remove entire experts considered less important based on their output over a calibration dataset. However, it is easy for them to accidentally discard important knowledge, leading to sharp accuracy drop (Chen et al., 2025). In contrast, expert merging methods attempt to combine similar experts rather than removing them entirely, typically via expert clustering (Chen et al., 2025) or using low-rank approximations (Li et al., 2025). Although these methods generally outperform expert dropping, they still experience significant accuracy degradation, with performance drops of over 20% on the MMLU benchmark (Chen et al., 2025; Li et al., 2025) as demonstrated in Figure 1.

The non-negligible performance degradation indicates a deeper tension of MoE compression: experts often contain a mix of shared knowledge (e.g., weights that are crucial for general language

Figure 1: (a): Accuracy of different MoE models on MMLU benchmark under 50%compression ratio with various expert compression methods, among which PuzzleMoE achieves the best accuracy. (b): Comparison of different expert compression methods, among which PuzzleMoE effectively and efficiently retains MoE models performance after compression with task-agnostic design.

modeling) and expert-specialized knowledge (e.g., parameters that handle particular inputs, domains, or linguistic patterns). Existing compression methods, whether dropping experts or merging full expert weights, risk discarding one or both types: dropping may eliminate critical specialization capabilities, while coarse-grained merging may hurt the distinctions between experts. As such, effective MoE compression must consider both preserving shared knowledge across experts while retaining expert specialization, which remain very challenging to achieve simultaneously. Moreover, many methods are not task-agnostic in design (Lu et al., 2024; Lee et al., 2025b) and require extensive offline compression time, as explored in Section 4.4. This highlights the need for an effective MoE merging approach that delivers both strong accuracy and practical compression efficiency.

In this paper, we propose PuzzleMoE, a training-free MoE compression method that achieves both high accuracy and efficient compression. First, we introduce a novel sparse expert merging algorithm that merges experts selectively at fine-grained weight entry level. Specifically, PuzzleMoE constructs two complementary masks: (1) An **entry-wise similarity mask** that identifies expert weights with high entry-wise similarity between expert pairs, which aims to identify the shared knowledge between experts; and (2) An **activation-weight saliency mask** that identifies weights critical to each expert's unique behavior, which ensures the unique knowledge of each expert can be reserved. By exploiting this dual-mask design, PuzzleMoE selectively merges only redundant parameters at fine-grained entry level while preserving expert specialization.

While entry-wise merging improves the preservation of shared and unique weights, it introduces the challenge of storing binary masks and sign bits per merged expert, which adds nontrivial overhead at scale. To overcome this, we introduce a bit-packed encoding scheme that reuses underutilized floating point bit representation. Specifically, we observe that the exponent field of expert weights often occupies a narrow range during inference, leaving multiple bits underutilized. We leverage these bits to embed binary masks and sign bits directly into the weight tensors, eliminating the need for auxiliary metadata storage. To support this format during inference, we design a custom CUDA kernel that decodes masks on the fly, enabling fast and memory-efficient execution of PuzzleMoE. Our contributions are summarized as follows:

- We propose PuzzleMoE, a sparse expert merging method that constructs entry-wise masks based on weight similarity and activation saliency and selectively merges expert in fine-grained ways to effectively preserve both shared knowledge and expert specialization.

- We design a bit-efficient encoding scheme that embeds masks and signs directly into weights, enabling metadata-free MoE inference with a lightweight custom CUDA kernel.

- We demonstrate PuzzleMoE's effectiveness across four MoE models and seven benchmarks. Specifically, it achieves up to 16.7% higher accuracy than prior methods under 50% compression ratio, along with $45\times$ faster compression and $1.28\times$ inference speedup for Mixtral-8x7B.

## 2 RELATED WORKS

Quantization and pruning are two widely adopted techniques for model compression. Recent research (Chen et al., 2025; Gu et al., 2025; Huang et al., 2025; Duanmu et al., 2025; Hu et al., 2025; Li et al., 2025; Lu et al., 2024) has explored their application to MoE models. Quantization techniques (Frantar et al., 2023; Lin et al., 2024) reduce memory footprint by exploiting per-weight redundancy, lowering weight precision to 4 bits. While quantization has achieved notable compression ratios with minimal impact on accuracy, the performance of pruning (including expert dropping and merging) within MoE models remains suboptimal. Specifically, pruning at a 50% sparsity ratio with existing methods results in a significant decline in accuracy (e.g. over 18.7% of accuracy drop on MMLU benchmark for Mixtral-8x7B) (Chen et al., 2025; Gu et al., 2025; Li et al., 2025; Lu et al., 2024), indicating that MoE model pruning still requires substantial advancement.

**Expert Dropping.** Expert dropping methods reduce MoE model size by eliminating entire expert modules considered unimportant. NAEE (Lu et al., 2024) performs an exhaustive search to determine which experts to retain, while STUN (Lee et al., 2025b) accelerates this process by reducing the selection complexity to a constant $O(1)$ by leveraging a latent structure between experts based on behavior similarity. These expert dropping methods suffer from a significant drawback: different downstream tasks require different selected calibration datasets. For the commonsense benchmarks, NAEE uses C4 for calibration, while for math tasks, it uses MATH dataset for calibration. MoE-I$^2$ (Yang et al., 2024a) performs inter-expert pruning followed by intra-expert low-rank decomposition. However, it requires finetuning to recover the performance of the compressed model.

**Expert Merging.** To better preserve accuracy, several recent works have proposed merging similar experts rather than dropping them. Methods like HC-SMoE (Chen et al., 2025) use hierarchical clustering based on expert output similarity to identify and combine experts, while MC-SMoE (Li et al., 2024), D2 (Gu et al., 2025), and Sub-MoE (Li et al., 2025) adopt multi-stage merging pipelines, e.g., first merging experts based on similarities, and then adding low-rank matrices to approximate the residual information. While these methods generally outperform expert dropping in accuracy, they often require complex procedures like SVD decomposition. In addition, they rely on coarse-grained expert merging, which risks hurting the distinctions between specialized experts. Different from existing methods, PuzzleMoE focuses on fine-grained entry-wise merging while being training-free and performing merging in a single pass.

**Model Merging in Dense Models.** Model merging is a promising approach for consolidating the capabilities of multiple pretrained and fine-tuned models into a single unified model (Yang et al., 2024b). Recent methods (He et al., 2025b; Zhao et al., 2025) have explored sparse merging, which selectively merges important parameters while avoiding the destructive effects of naive weight averaging. Different from those work, PuzzleMoE introduces sparse merging principles to MoE models: it performs entry-wise merging with efficient inference through our bit-packing optimization. To our knowledge, PuzzleMoE is the first method that explores sparse merging for MoE model compression at fine-grained entry level.

## 3 METHOD

In this section, we introduce our design of PuzzleMoE, which achieves efficient fine-grained sparse expert merging through a deliberately designed pairwise dual-mask merging algorithm and a system co-design with bit-level packing technique. These two designs can effectively maintain MoE models' performance after compression while minimizing the masking overhead at the inference stage. We detail these two designs in Section 3.1 and Section 3.2, respectively.

### 3.1 PAIRWISE DUAL-MASK EXPERT MERGING

Consider two weight matrices $\mathbf{W}_i$ and $\mathbf{W}_j \in \mathcal{R}^{d \times h}$, which correspond to experts $\mathbf{E}_i$ and $\mathbf{E}_j$ from an MoE layer of a model with $N$ experts $\varepsilon = \{\mathbf{E}_1, \mathbf{E}_2, \ldots, \mathbf{E}_N\}$, our goal is to construct a merged expert $\mathbf{W}_{merged} \in \mathcal{R}^{d \times h}$ that preserves shared information among experts' weights while retaining expert-specific, high-saliency parameters. To this end, we propose a dual-mask element-wise merging strategy to obtain $\mathbf{W}_{merged}$: A *similarity-based mask* that identifies shared weight entries, and a *saliency-based mask* used to preserves divergent yet important weight entries. Additionally,

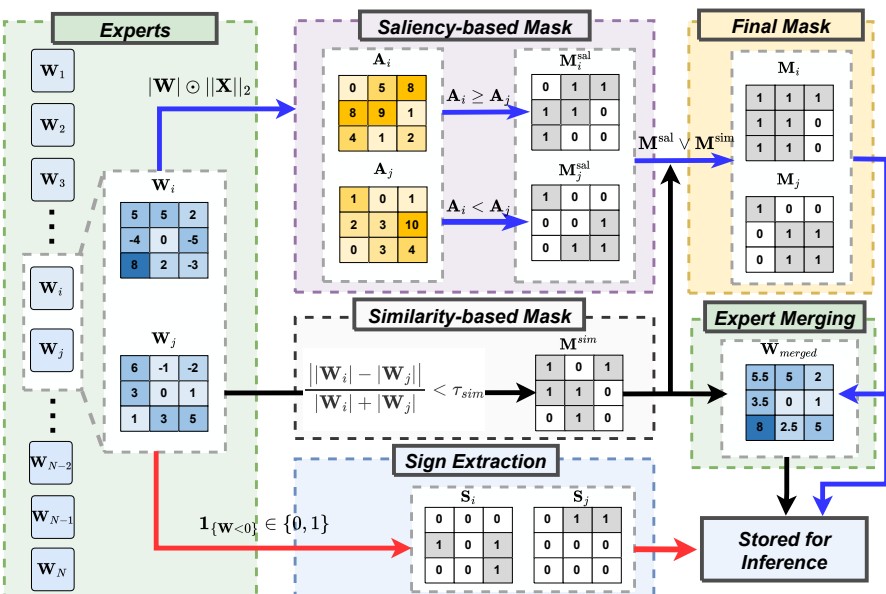

Figure 2: Procedure of sparse expert merging algorithm.

we store the original signs of each expert's weights and reapply them to $\mathbf{W}_{merged}$ during inference. The overall algorithm design is illustrated in Figure 2.

**Pair-wise Merging.** We adopt a pairwise merging strategy for efficiency and tractability. The rationale is that when attempting to merge $k \geq 3$ experts jointly, each weight index admits $(2^k - 1)$ choices, yielding a combinatorial masking problem whose difficulty scales exponentially in $k$ and is intractable in practice, especially for the modern MoE models with a large number of experts. By contrast, pairwise merging enables closed-form mask construction with linear-time complexity and compact encoding. Moreover, pairwise merging performs better than the combinatorial merging under the same merging sparsity as discussed in Section 5.

**Similarity-based Mask.** Given two experts weights $\mathbf{W}_i$ and $\mathbf{W}_j$, we measure their per-entry magnitude similarity using a symmetric percent difference (Miller, 2011):

$$\boldsymbol{\Delta} := \frac{||\mathbf{W}_i| - |\mathbf{W}_j||}{|\mathbf{W}_i| + |\mathbf{W}_j|}, \tag{1}$$

where smaller values indicate greater similarity between two weight entries. Eq.1 cleanly identifies entries in two experts' weights that are similar in magnitude regardless of direction to avoid spurious penalties from opposite signs and minimize distortion in the reconstructed weights when signs are restored at inference. After that, the similarity-based mask between two experts can be defined as:

$$\mathbf{M}^{\mathrm{sim}} := \mathbf{1}_{\{\boldsymbol{\Delta} \leq \tau_{\mathrm{sim}}\}} \in \{0, 1\}^{d \times h}, \tag{2}$$

where $\tau_{\mathrm{sim}} \in [0, 1]$ is the pre-defined similarity threshold. $\mathbf{M}^{\mathrm{sim}}$ is used to identify experts' entries with comparable magnitude so that they can be safely aggregated. We also analyze the reason why different experts have weight similarity in Appendix B.2 to support this design. As discussed, we also need to store each expert's sign pattern and reapply it to $\mathbf{W}_{merged}$ at inference, ensuring that shared merged entries reconstruct the original signed weights with minimal distortion:

$$\mathbf{S}_i := \mathbf{1}_{\{\mathbf{W}_i < 0\}} \in \{0, 1\}^{d \times h}, \qquad \mathbf{S}_j := \mathbf{1}_{\{\mathbf{W}_j < 0\}} \in \{0, 1\}^{d \times h}. \tag{3}$$

**Saliency-based Mask.** To decide which expert's entries to preserve, we extend the idea of quantifying the importance of weights by combining the magnitude of weights with the saliency of input activations in dense models (Sun et al., 2024) to MoE experts:

$$\mathbf{A}_i = |\mathbf{W}_i| \odot ||\mathbf{X}_i||_2, \qquad \mathbf{A}_j = |\mathbf{W}_j| \odot ||\mathbf{X}_j||_2, \tag{4}$$

where $\mathbf{X}$ represents a sample of input activations to a certain expert. Therefore, the complementary saliency masks can be obtained as:

$$\mathbf{M}_i^{\mathrm{sal}} := \mathbf{1}_{\{\mathbf{A}_i \geq \mathbf{A}_j\}} \in \{0, 1\}^{d \times h}, \qquad \mathbf{M}_j^{\mathrm{sal}} := \mathbf{1} - \mathbf{M}_i^{\mathrm{sal}}. \tag{5}$$

These two masks indicate which expert has the more important weight to be reserved at each entry position of the final $\mathbf{W}_{merged}$.

**Sparse Expert Merging.** To achieve the preservation of critical importance weights between two experts while minimizing the redundancy within them, we average the magnitudes of entries whose values are similar in terms of $\mathbf{M}_{\text{sim}}$, whereas dissimilar entries are selected from the more salient expert using the saliency masks. This sparse merging process can be expressed as:

$$\mathbf{M}_i = \mathbf{M}_i^{\text{sal}} \vee \mathbf{M}^{\text{sim}}, \qquad \mathbf{M}_j = \mathbf{M}_j^{\text{sal}} \vee \mathbf{M}^{\text{sim}}, \tag{6}$$

$$\mathbf{W}_{\text{merged}} = \mathbf{M}^{\text{sim}} \odot \frac{|\mathbf{W}_i| + |\mathbf{W}_j|}{2} + (\mathbf{1} - \mathbf{M}^{\text{sim}}) \odot (\mathbf{M}_i^{\text{sal}} \odot |\mathbf{W}_i| + \mathbf{M}_j^{\text{sal}} \odot |\mathbf{W}_j|). \tag{7}$$

The merging process is done offline, and $\mathbf{W}_{\text{merged}}$, $\mathbf{M}_i$, $\mathbf{M}_j$, $\mathbf{S}_i$, $\mathbf{S}_j$ are stored. At the inference stage, once an expert is activated, its weights are reconstructed element-wise as

$$\widehat{\mathbf{W}}_i = (-1)^{\mathbf{S}_i} \odot \mathbf{M}_i \odot \mathbf{W}_{\text{merged}}. \tag{8}$$

Unlike prior search-based methods, our merging process requires only a single forward pass to compute expert saliency scores using the Wanda (Sun et al., 2024) metric, completely obviating the need for an expensive search procedure in NAEE (Lu et al., 2024). The detailed comparison of offline compression can be found in Section 4.4.

**How to select the pairwise experts grouping strategy?** We propose two strategies for grouping experts into pairs for merging: (1) random grouping, (2) search-based grouping. In practice, we find that random grouping performs sufficiently well, as discussed in Section 5. Therefore, we adopt random grouping as the default strategy for pairwise expert combination. Additional details and sensitivity analyses on hyperparameter configurations are provided in Section 5.

### 3.2 EFFICIENT INFERENCE WITH BIT-PACKING

The sparse expert merging process transforms each pair of experts into a merged expert, 2 sign matrices, and 2 corresponding binary masks, which present a significant challenge to achieving efficient inference. As revealed by Lasby et al. (2025), employing Compressed Sparse Row format to store a tensor with 50% unstructured sparsity does not result in any memory savings. This inefficiency arises primarily from the considerable overhead associated with storing index information. Furthermore, existing expert calculation relies on dense matrix multiplications, which is highly optimized on modern GPUs and they don't support calculation with masks. The need to dynamically fetch the appropriate mask during inference introduces considerable latency, particularly in the scheduling and memory access of matrix multiplication kernels. To overcome these limitations, we propose a solution combining efficient bit-level packing with a high-performance GEMV CUDA kernel.

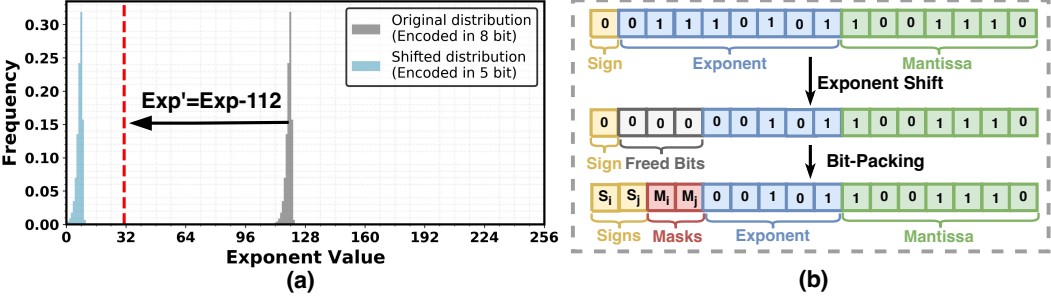

Figure 3: Illustration of the mask packing procedure. (a): the distribution of Bfloat16 weight exponents in Mixtral-8x7B. After shifting, the exponents can be encoded in 5 bits. (b): bit-level organization of masks and signs within the packed Bfloat16 format.

#### 3.2.1 OBSERVATION

The Bfloat16 data format, which allocates 1 sign, 8 exponent, and 7 mantissa bits, is standard for LLM inference. Recent analyses (Su et al., 2024; Zhang et al., 2025; Lee et al., 2025a) reveal significant underutilization of the 8-bit exponent range for dense LLMs in Bfloat16 format. We

found that this also applies for MoE models. As illustrated in Figure 3, for the Mixtral-8x7B (Jiang et al., 2024) model, the exponent values of expert weights are predominantly concentrated within a narrow range of 112 to 128. We also observe this in other MoE models as shown in Appendix B.1.

### 3.2.2 SYSTEM CO-DESIGN

Leveraging the concentrated exponent distribution, we apply a fixed shift to map all exponents into a 5-bit range, as illustrated in Figure 3. Specifically, any exponent smaller than 112 is rounded up to 112, and then all exponents are shifted down by 112, resulting in values that fall within the range of 0 to 31. As

Table 1: Perplexity results on Wikitext2 before and after bit-packing.

| Model | Before Pack | After Pack |
|---|---|---|
| Mixtral-8x7B | 4.37 | 4.37 |
| Deepseek-MoE | 6.88 | 6.88 |

detailed in Table 1, this transformation incurs no perplexity degradation for the existing large MoE models because crafting an FP16 model from a Bfloat16 model follows similar procedure for the exponent part. This shift operation frees 3 bits within the original 8-bit exponent field, which can be used to pack the binary mask bits and a sign bit. Hence, the resultant $\mathbf{W}_{merged}$ is stored in a standard Bfloat16 data format, effectively embedding the masks and signs without additional storage.

---

**Algorithm 1** PuzzleMoE weight decoding

---

**Input**: W: Merged weight value, expert_pos: 0 or 1 expert in the merge group
**Output**: W_decoded

1: mask_bit ← (W ≫ (13 - expert_pos)) & 1      ▷ Extract the packed mask bit for the weight
2: **if** mask_bit==0:
3:      W_decoded ← 0      ▷ The weight value is 0 if it is marked as pruned
4: **else**:
5:      sign_bit ← (W ≫ (15 - expert_pos)) & 1      ▷ Extract the packed sign bit for the weight
6:      exp ← (W & $0x0F80$) + (112 ≪ 7)      ▷ Rebuild the bfloat16 exponent
7:      W_decoded ← (sign_bit ≪ 15) | exp | (W & $0x007F$)      ▷ Reconstruct the weight value
8: W_decoded.view(bfloat16)

---

We develop a specialized GEMV CUDA kernel that incorporates on-the-fly decoding to realize the benefit of the above bit packing idea. As detailed in Algorithm 1, each weight $W[i, j]$ is dynamically decoded from its packed format immediately prior to its use in the multiplication $X[i, j] \times W[i, j]$. The efficiency of this approach hinges on its synergistic design. The decoding logic is a computationally trivial, in-place operation that piggybacks on the kernel's existing data-loading path, which is already highly optimized for maximum memory throughput via techniques like warp-level scheduling and coalesced memory access. We eliminate the need for a separate materialization of the decoded matrix in memory, thereby avoiding significant latency and memory access overhead.

## 4 EXPERIMENTS

### 4.1 EXPERIMENTAL SETTINGS

**Models and Baselines.** We conduct a comprehensive evaluation of PuzzleMoE on four state-of-the-art MoE models: Mixtral-8x7B (Jiang et al., 2024), Deepseek-MoE (Dai et al., 2024), Qwen1.5-MoE-A2.7B (Team, 2024), and Qwen3-MoE-30B-A3B (Yang et al., 2025). We follow prior work to evaluate two distinct compression ratios of 25% and 50%, reducing the number of experts to 75% and 50% of the original count, while keeping the other modules unchanged. We compare PuzzleMoE against existing MoE compression methods, including expert dropping methods NAEE (Lu et al., 2024), STUN (Lee et al., 2025b) and expert merging methods D2 (Gu et al., 2025), HC-SMoE (Chen et al., 2025), Sub-MoE (Li et al., 2025). We also compare with LLM pruning algorithm Wanda (Sun et al., 2024), whose 2:4 semi-structured sparsity is applied exclusively to the experts to ensure a fair comparison [1].

---

[1] This is different from the setting in NAEE. In NAEE, the Wanda pruning is applied uniformly across all linear modules in the MoE model, which introduces an unfair comparison. Specifically, the attention module is more sensitive to pruning than the experts module.

Table 2: Zero-shot comparison of Mixtral-8x7B, Deepseek-MoE, Qwen1.5-MoE, and Qwen3-MoE under 25% and 50% sparsity.

| Method | Sparsity | Wiki | ARC-c | ARC-e | Hella | Piqa | BoolQ | Wino | MMLU | Avg |
|---|---|---|---|---|---|---|---|---|---|---|
| **Mixtral-8x7B-v0.1** | | | | | | | | | | |
| Vanilla | 0% | 3.84 | 56.7 | 84.1 | 64.9 | 82.4 | 85.4 | 77.2 | 67.9 | 74.1 |
| NAEE | 25% | 5.01 | 52.0 | 82.0 | 61.9 | 80.9 | 84.0 | 75.1 | 58.1 | 70.6 |
| STUN | 25% | - | 52.7 | 81.8 | 60.8 | - | 83.1 | 72.7 | 63.3 | - |
| D2 | 20% | 4.65 | 51.0 | 80.0 | 61.0 | 81.0 | - | 75.0 | - | - |
| HC-SMoE | 25% | 5.31 | 50.3 | 79.3 | 61.3 | 80.7 | 84.9 | 75.4 | 59.4 | 70.2 |
| Sub-MoE | 25% | 5.16 | 49.0 | 80.0 | 62.0 | - | **86.0** | 75.0 | 59.0 | - |
| PuzzleMoE | 25% | **4.10** | **55.3** | **83.2** | **64.2** | **82.1** | 85.4 | **75.5** | **66.8** | **73.2**$_{\pm0.2}$ |
| NAEE | 50% | 6.49 | 48.1 | 78.5 | 57.8 | 79.1 | 81.0 | 73.0 | 47.3 | 66.4 |
| D2 | 40% | 5.28 | 47.0 | 78.0 | 57.0 | 78.0 | - | 73.0 | - | - |
| HC-SMoE | 50% | 7.65 | 41.1 | 72.0 | 55.5 | 76.0 | 80.8 | 72.1 | 49.0 | 63.8 |
| Wanda | 2:4 | 5.89 | 48.3 | 78.8 | 58.7 | 79.5 | 79.2 | 74.5 | 62.0 | 68.7 |
| Wanda | 50% | 4.68 | 53.1 | 81.8 | 62.4 | 81.4 | 85.9 | 76.6 | 65.8 | 72.4 |
| Sub-MoE | 50% | 6.97 | 45.0 | 75.0 | 57.0 | - | 84.0 | 72.0 | 48.0 | - |
| PuzzleMoE | 50% | **4.36** | **53.8** | **82.4** | **63.3** | **81.7** | **85.3** | **75.8** | **65.7** | **72.6**$_{\pm0.2}$ |
| **Deepseek-MoE-16b** | | | | | | | | | | |
| Vanilla | 0% | 6.51 | 44.6 | 75.9 | 58.1 | 78.8 | 72.8 | 70.1 | 37.8 | 62.6 |
| D2 | 20% | 6.84 | 41.0 | 74.0 | 55.0 | 76.0 | - | 69.0 | - | - |
| HC-SMoE | 25% | 24.48 | 36.7 | 65.1 | 44.3 | 73.1 | 66.4 | 65.7 | 24.5 | 53.7 |
| Sub-MoE | 25% | 8.48 | 40.0 | 72.0 | 54.0 | - | 73.0 | 70.0 | 27.0 | - |
| PuzzleMoE | 25% | **6.68** | **44.0** | **75.7** | **57.2** | **78.7** | **73.1** | **70.6** | **37.2** | **62.4**$_{\pm0.3}$ |
| D2 | 40% | 7.93 | 36.0 | 69.0 | 45.0 | 72.0 | - | 65.0 | - | - |
| Wanda | 2:4 | 8.46 | 37.4 | 71.2 | 51.4 | 76.2 | **75.9** | 69.2 | 31.0 | 58.9 |
| Wanda | 50% | 7.18 | 41.4 | 74.8 | 54.8 | 77.9 | 76.3 | 70.6 | 35.4 | 61.6 |
| HC-SMoE | 50% | 89.94 | 22.3 | 41.9 | 31.2 | 62.3 | 62.3 | 55.3 | 23.0 | 42.6 |
| Sub-MoE | 50% | 13.71 | 32.0 | 63.0 | 44.0 | - | 68.0 | 65.0 | 22.0 | - |
| PuzzleMoE | 50% | **6.88** | **43.0** | **75.2** | **56.3** | **78.4** | 74.5 | **70.3** | **36.9** | **62.1**$_{\pm0.4}$ |
| **Qwen1.5-MoE-A2.7B** | | | | | | | | | | |
| Vanilla | 0% | 7.22 | 41.0 | 73.2 | 58.0 | 80.0 | 79.5 | 68.9 | 61.0 | 65.9 |
| HC-SMoE | 25% | 11.28 | 34.8 | 67.5 | 50.4 | 74.1 | 74.2 | 66.1 | 51.0 | 59.7 |
| PuzzleMoE | 25% | **7.37** | **40.9** | **73.4** | **57.3** | **79.7** | 79.2 | **69.6** | **60.4** | **65.8**$_{\pm0.2}$ |
| Wanda | 2:4 | 8.81 | 38.7 | 72.2 | 52.6 | 77.5 | 76.2 | 67.8 | 55.8 | 63.0 |
| HC-SMoE | 50% | 78.04 | 23.9 | 41.1 | 31.2 | 60.9 | 56.6 | 56.1 | 23.2 | 41.9 |
| PuzzleMoE | 50% | **7.55** | **40.7** | **73.5** | **56.5** | **79.4** | **78.6** | **69.4** | **60.0** | **65.4**$_{\pm0.2}$ |
| **Qwen3-MoE-30B-A3B** | | | | | | | | | | |
| Vanilla | 0% | 8.70 | 52.7 | 79.3 | 59.5 | 79.6 | 88.7 | 70.4 | 77.8 | 72.6 |
| Sub-MoE | 25% | 13.59 | 44.0 | 70.0 | 47.0 | - | 86.0 | 66.0 | 65.0 | - |
| PuzzleMoE | 25% | **9.08** | **51.6** | **78.9** | **58.3** | **79.3** | **88.2** | **70.4** | **76.6** | **71.9**$_{\pm0.3}$ |
| Wanda | 2:4 | 11.77 | 48.2 | 76.1 | 51.1 | 76.3 | 88.0 | **70.5** | 72.1 | 68.9 |
| Sub-MoE | 50% | 21.05 | 40.0 | 69.0 | 41.0 | - | 84.0 | 63.0 | 56.0 | - |
| PuzzleMoE | 50% | **9.50** | **51.0** | **78.5** | **57.1** | **78.9** | **88.0** | 70.1 | **75.1** | **71.2**$_{\pm0.4}$ |

**Benchmarks and Evaluation.** We evaluate the compressed models on language modeling perplexity and zero-shot task performance. Language modeling capabilities are assessed on the WikiText-2 (Merity et al., 2016) with a sequence length of 2048. For downstream tasks, we evaluate zero-shot accuracy across seven common benchmarks: ARC-c (Clark et al., 2018), ARC-e (Clark et al., 2018), HellaSwag (Zellers et al., 2019), PIQA (Bisk et al., 2019), BoolQ (Clark et al., 2019), Winogrande (Sakaguchi et al., 2019), and MMLU (Hendrycks et al., 2021). For all experiments, the calibration dataset has 128 samples, each with a sequence length of 2048 drawn from the C4 dataset (Raffel et al., 2023). We set the similarity threshold $\tau_{\text{sim}} = 0.4$ fixed for all models and tasks, and evaluated 16 different random seeds for expert combination. The results in Table 2, Table 3, and Table 4 for PuzzleMoE are the average result. The detailed results for each seed are shown in Appendix B.6.

## 4.2 PERFORMANCE ON GENERAL TASKS

From Table 2, we observe that for Mixtral-8x7B, PuzzleMoE achieves an average accuracy of 73.2% at 25% sparsity and 72.6% at 50% sparsity, substantially outperforming other baselines. For DeepSeek-MoE, PuzzleMoE incurs only minimal accuracy drops of 0.2% and 0.5% at 25% and 50% sparsity, respectively. Similarly, for Qwen1.5-MoE, the degradation is limited to 0.1% and 0.5%, while for Qwen3-MoE, the drops are 0.7% and 1.4% under 25% and 50% sparsity. These

results highlight the effectiveness of PuzzleMoE on reserving MoE models performance after expert merging, whose superior performance arises from its fine-grained sparse expert merging strategy.

## 4.3 PERFORMANCE ON DOMAIN-SPECIFIC TASKS

**Math Reasoning Tasks.** We evaluate Puzzle-MoE on domain-specific mathematical reasoning tasks to assess its ability to preserve reasoning performance under sparsity, as shown in Table 3. PuzzleMoE consistently achieves the best results among all methods, with accuracies of 55.4% and 51.7% at 25% and 50% sparsity, respectively. Moreover, the effectiveness of NAEE is strongly influenced by the choice of calibration data, performing better with the Math dataset than with C4. In contrast, PuzzleMoE remains robust regardless of calibration as shown in Section 5.

Table 3: 8-shot GSM8K accuracy on Mixtral-8x7B.

| Sparsity | Method | Calib. data | GSM8K |
|---|---|---|---|
| 0% | - | - | 57.6 |
| 25% | NAEE | C4 | 41.5 |
| | NAEE | Math | 48.7 |
| | PuzzleMoE | C4 | **55.4** |
| 50% | NAEE | C4 | 28.6 |
| | NAEE | Math | 38.8 |
| | Wanda | C4 | 32.2 |
| | PuzzleMoE | C4 | **51.7** |

We also evaluate the Qwen3-MoE-30B-A3B model (Yang et al., 2025) on the more challenging reasoning benchmarks (generation-intensive) with results reported in Table 4. Notably, under 25% sparsity, PuzzleMoE effectively reserves the model's reasoning ability after compression, retaining 99%, 92%, and 84%

Table 4: Performance on math reasoning benchmarks.

| Method | Math-500 | AIME24 | AIME25 |
|---|---|---|---|
| Baseline (0%) | 97.2 | 83.3 | 72.9 |
| HC-SMoE (25%) | 24.6 | 0.0 | 0.0 |
| PuzzleMoE (25%) | **96.2** | **71.1** | **61.5** |

of accuracy for the three benchamrks, respectively. In sharp contrast, HC-SMoE collapses to 24.6, 0.0, and 0.0 under the same sparsity. This highlights the effectiveness of PuzzleMoE in preserving the reasoning capability of MoE model compared to the previous method.

**Coding Tasks.** We further investigate the knowledge preservation ability of PuzzleMoE on coding tasks. Specifically, we compare different expert merging algorithms on Qwen1.5-MoE-A2.7B and evaluate them on HumanEval (Chen et al., 2021). As shown in Table 5, PuzzleMoE preserves code-generation performance much better than HC-SMoE under the same sparsity levels: at 25% sparsity, PuzzleMoE attains a Pass@1 of 45.1 with only a small drop from the 49.4 dense baseline, while HC-SMoE collapses to 11.0; at 50% sparsity,

Table 5: Performance on HumanEval.

| Model | HumanEval Pass@1 |
|---|---|
| Baseline (0%) | 49.4 |
| HC-SMoE (25%) | 11.0 |
| PuzzleMoE (25%) | **45.1** |
| HC-SMoE (50%) | 0.0 |
| PuzzleMoE (50%) | **39.6** |

PuzzleMoE still achieves 39.6, whereas HC-SMoE completely fails with 0.0 Pass@1. These results demonstrate that PuzzleMoE maintains strong coding ability even at high sparsity, while prior MoE merging methods severely degrade model reliability.

## 4.4 EFFICIENCY ANALYSIS

**Compression Cost.** We measure the time required to compress the MoE models to 50% expert sparsity on 2 A100-80G GPUs for PuzzleMoE, D2, HC-SMoE, and NAEE. As shown in Figure 4, PuzzleMoE only takes 2 minutes for compressing Mixtral-8x7B, while D2 takes 55 minutes due to the heavy computation cost of SVD operation. For Deepseek-MoE, PuzzleMoE takes 10 minutes for compression, indicating the efficiency of PuzzleMoE on MoE models with a large number of experts. Note that NAEE's reliance on exhaustive search makes the compression time quite expensive. For instance, applying it to DeepSeek-MoE with 64 experts requires $10^{18}$ on the order of forward passes, rendering it infeasible in practice.

**Memory Usage and Inference Speedup.** We evaluate the memory usage and inference speedup of Mixtral-8x7B and Qwen3-MoE-30B-A3B under 50% compression ratio as shown in Figure 4. The prefill and decode lengths are set to 1024 and 512, respectively. For Mixtral-8x7B, the full model necessitates two A100-80G GPUs for inference, whereas the compressed model can be deployed on a single A100-80G GPU. Similarly, for Qwen3-MoE, which is evaluated on A100-40G GPUs, the

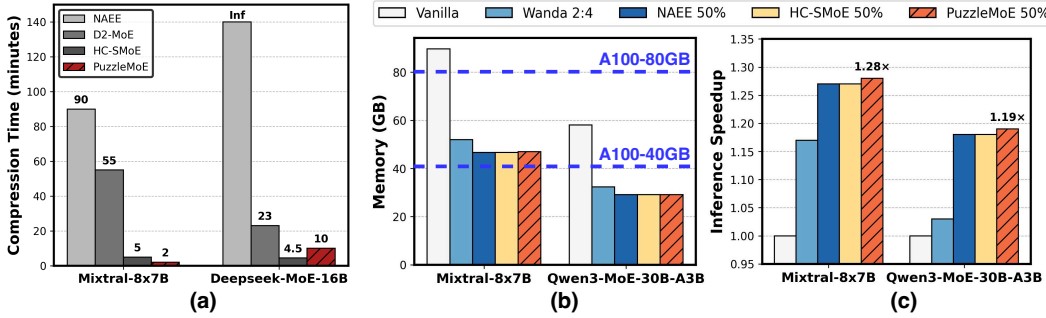

Figure 4: System performance for different MoE models and tasks. (a): Compression time comparison across Mixtral and Deepseek-MoE. (b): Memory usage during inference for Mixtral-8x7B and Qwen3-MoE. (c): Inference speedup comparison for Mixtral-8x7B and Qwen3-MoE.

full model requires two GPUs for inference, while the compressed model is capable of inference on a single GPU. Moreover, PuzzleMoE achieves an inference speedup of $1.28\times$ for Mixtral-8x7B and $1.19\times$ for Qwen3-MoE, which are marginally better than the existing compression methods and with great accuracy gains as shown in Table 2. This speedup mainly comes from two factors: (1) Eliminated cross-GPU communication: Moving from 2 GPUs to 1 removes expensive cross-GPU communication and synchronization overhead. (2) Kernel-level optimizations: Our custom GEMV kernel is tailored to the merged-expert structure, which further reduces overhead and improves efficiency on a single device. Therefore, PuzzleMoE attains efficiency gains comparable to existing compression methods while providing markedly better accuracy, making it a more balanced approach for deploying MoE models.

## 5 ABLATION STUDY

**Impact of Calibration Datasets.** We futher explore the impact of different calibration datasets on PuzzleMoE, and the results are illustrated in Table 6. We can observe that the using C4 or MATH as the calibration dataset doesn't lead to a big variance in the performance on different downstream tasks, emphasizing the robustness of PuzzleMoE.

**Expert Grouping Strategy.** We compare our random pairwise expert grouping strategy with an evolutionary search-based pairwise grouping strategy. As shown in Table 8, the search-based grouping strategy only leads to a slight performance increase compared with the default random one, indicating that PuzzleMoE's effectiveness is hardly influenced by the pairwise grouping strategy.

**Number of Experts to Merge.** We ablate the number of experts to merge in PuzzleMoE. As shown in Table 7, at a 50% sparsity level, increasing the merge group size from 2 to 3 experts leads to degraded performance, as reflected by higher perplexity. Moreover, merging three experts introduces notable hardware inefficiencies: encoding the selection masks and signs for three experts requires 5 bits, which exceeds the 3 redundant bits available in the Bfloat16 format, making the design impractical.

Table 6: PuzzleMoE with different calibration datasets for Mixtral-8x7B.

| Calib. data | GSM8K | Avg Accuracy |
|---|---|---|
| C4 | 51.7 | 72.6 |
| Math | 51.7 | 72.5 |

Table 7: Impact of different numbers of experts to merge with 50% sparsity.

| Model | Merge Number | Wiki |
|---|---|---|
| Mixtral-8x7B | 2 | 4.36 |
| | 3 | 5.22 |
| Deepseek-MoE | 2 | 6.88 |
| | 3 | 7.75 |

Table 8: Impact of different expert grouping strategies with 50% sparsity.

| Model | Method | Wiki | Avg Acc |
|---|---|---|---|
| Mixtral-8x7B | Random | 4.36±0.01 | 72.6±0.2 |
| | Searched | 4.35 | 72.9 |
| Deepseek-MoE | Random | 6.88±0.01 | 62.1±0.3 |
| | Searched | 6.86 | 62.4 |

**Similarity Threshold** $\tau_{\text{sim}}$**.** We report perplexity results for PuzzleMoE with different values of $\tau_{\text{sim}}$ as shown in Figure 5. It is clear that small values underuse magnitude similarity, while large

values merge too aggressively and leads a substantially higher loss across the two experts. Values of $\tau_{\text{sim}}$ within the range of 0.3 to 0.5 yield the best performance; therefore, we fix $\tau_{\text{sim}} = 0.4$ across models.

**Combining PuzzleMoE with Quantization.** We also investigated the compatibility of PuzzleMoE with quantization. Specifically, we apply a symmetric group quantization scheme to the resultant merged weights. As depicted in Figure 5, this combined approach achieves a substantial compression ratio of $4.8\times$, with a minimal accuracy drop of only 1.7% for Mixtral-8x7B and 1.0% for Deepseek-MoE compared with full models, demonstrating that our merging strategy can be effectively combined with quantization without significant performance loss. Further details of quantized PuzzleMoE are shown in Appendix B.3.1.

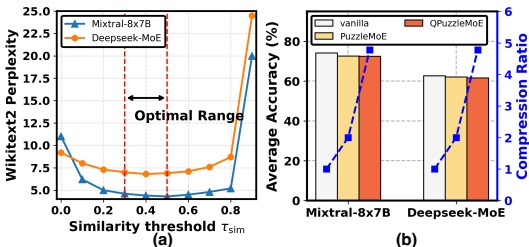

Figure 5: (a): Wikitext2 perplexity of Mixtral and Deepseek-MoE under different similarity thresholds. (b): Accuracy performance of combining PuzzleMoE with quantization.

## 6 CONCLUSION

We introduce PuzzleMoE, an efficient compression method for large Mixture-of-Experts models. Our approach utilizes a sparse expert merging strategy guided by dual masks, which enables model compression within minutes while effectively preserving downstream task performance. Furthermore, by incorporating bit-packed encoding, PuzzleMoE facilitates efficient decoding on GPUs, offering a practical solution for deploying large MoE models in real-world applications.

## REPRODUCIBILITY STATEMENT

We have taken extensive measures to ensure the reproducibility of the results presented in this paper. The experimental setup, including model configurations and hardware specifications, is provided in detail. We hope these efforts will enable others to replicate our work and contribute to further advancements in the field.

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

## A  LLM USAGE

Large language models (LLMs) were used solely to improve the writing of this paper, including grammar, clarity, and readability. They were not used for generating ideas, designing experiments, conducting analyses, or producing scientific content. All research contributions, technical claims, and conclusions are entirely the work of the authors.

## B  APPENDIX

### B.1  EXPONENT DISTRIBUTION VISUALIZATION OF DIFFERENT MoE MODELS

We provide a visualization of the exponent distribution of different MoE models in Figure 6.

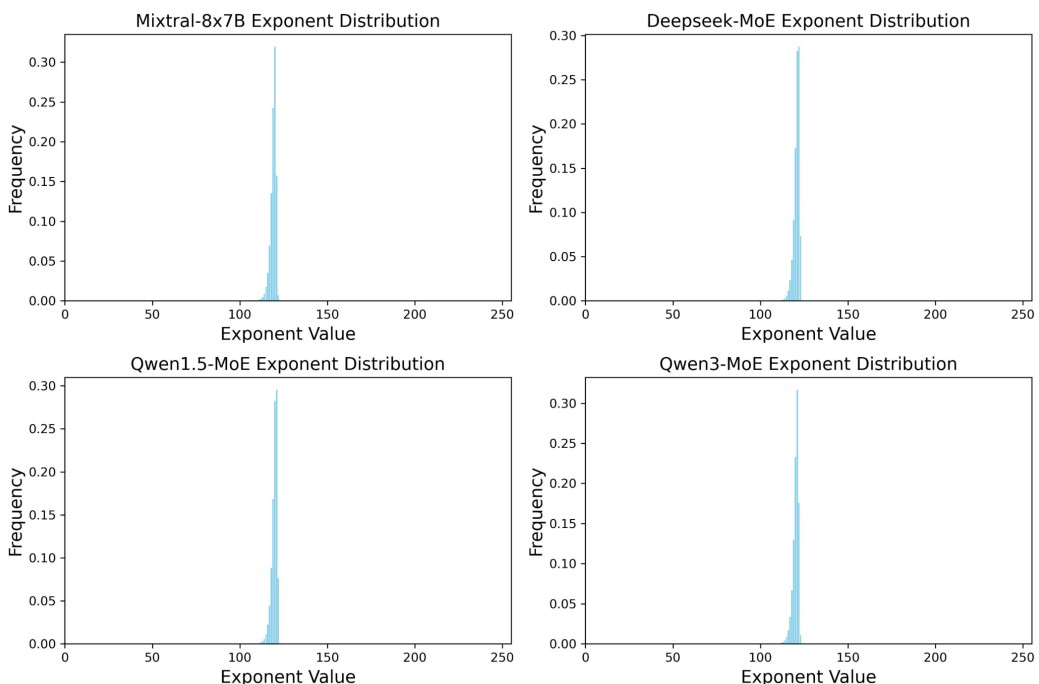

Figure 6: Exponent distribution of different MoE models.

### B.2  EXPLANATION FOR ELEMENT-WISE SIMILARITY IN MODEL WEIGHTS

We conduct an extended analysis of entry-level similarity in experts' weights of MoE models. Consistent with previous findings (Kim et al., 2024), the weight distribution of modern LLMs approximates a normal distribution with a zero mean. To quantify inter-expert similarity, we computed the Pearson correlation coefficient between tensors in different pairs of experts within each layer, then we averaged the result for the whole model. As shown in Table 9, for Qwen1.5-MoE and Deepseek-MoE, the correlation is negligible. In contrast, Mixtral-8x7B exhibits a substantially higher correlation, indicating stronger dependencies between its expert weights. Consequently, for Qwen1.5-MoE and Deepseek-MoE, the values within their respective tensors can be treated as independent variables.

Based on this setting, we can view the process of selecting 2 values $w_1, w_2$ of the corresponding positions of the 2 weights as a sampling process. We define and solve the problem of calculating the proportion of similar values as follows:

Table 9: Average pearson correlation of expert weights.

| Model | Average correlation |
|---|---|
| Mixtral-8x7B | 0.2612 |
| Deepseek-MoE | 0.0006 |
| Qwen1.5-MoE | 0.0000 |

We consider independent random variables

$$w_1 \sim N(0, \sigma_1^2), \quad w_2 \sim N(0, \sigma_2^2),$$

and define

$$A = |w_1|, \quad B = |w_2|.$$

We aim to compute

$$P\left(\frac{|A - B|}{A + B} < \tau_{\text{sim}}\right), \qquad 0 < \tau_{\text{sim}} < 1.$$

Since $A, B \geq 0$, we have

$$\frac{|A - B|}{A + B} < \tau_{\text{sim}} \iff |A - B| < \tau_{\text{sim}}(A + B) \iff \frac{1 - \tau_{\text{sim}}}{1 + \tau_{\text{sim}}} < \frac{A}{B} < \frac{1 + \tau_{\text{sim}}}{1 - \tau_{\text{sim}}}.$$

Define the ratio

$$R = \frac{A}{B} = \frac{|w_1|}{|w_2|}.$$

The densities of $A$ and $B$ (half-normal distributions) are given by

$$f_A(a) = \frac{\sqrt{2}}{\sigma_1 \sqrt{\pi}} e^{-a^2/(2\sigma_1^2)}, \quad a \geq 0,$$

$$f_B(b) = \frac{\sqrt{2}}{\sigma_2 \sqrt{\pi}} e^{-b^2/(2\sigma_2^2)}, \quad b \geq 0.$$

For $R = A/B$, its pdf is

$$f_R(r) = \int_0^\infty b\, f_A(rb)\, f_B(b)\, db = \frac{2\sigma_1 \sigma_2}{\pi\left(\sigma_1^2 + \sigma_2^2 r^2\right)}, \quad r \geq 0.$$

Let

$$a = \frac{1 - \tau_{\text{sim}}}{1 + \tau_{\text{sim}}}, \qquad b = \frac{1 + \tau_{\text{sim}}}{1 - \tau_{\text{sim}}}.$$

Then

$$P = \int_a^b f_R(r)\, dr = \frac{2\sigma_1 \sigma_2}{\pi} \int_a^b \frac{dr}{\sigma_1^2 + \sigma_2^2 r^2} = \frac{2}{\pi}\left[\arctan\left(\frac{\sigma_2 r}{\sigma_1}\right)\right]_a^b.$$

Hence, the final result is

$$P\left(\frac{|\,|w_1| - |w_2|\,|}{|w_1| + |w_2|} < \tau_{\text{sim}}\right) = \frac{2}{\pi}\left[\arctan\left(\frac{\sigma_2}{\sigma_1}\frac{1 + \tau_{\text{sim}}}{1 - \tau_{\text{sim}}}\right) - \arctan\left(\frac{\sigma_2}{\sigma_1}\frac{1 - \tau_{\text{sim}}}{1 + \tau_{\text{sim}}}\right)\right].$$

In the case $\sigma_1 \approx \sigma_2$, which typically holds for most expert weights in MoE models, we obtain

$$P\left(\frac{|\,|w_1| - |w_2|\,|}{|w_1| + |w_2|} < \tau_{\text{sim}}\right) = \frac{2}{\pi}\left[\arctan\left(\frac{1 + \tau_{\text{sim}}}{1 - \tau_{\text{sim}}}\right) - \arctan\left(\frac{1 - \tau_{\text{sim}}}{1 + \tau_{\text{sim}}}\right)\right].$$

The theoretical curve and empirical data from the three MoE models are plotted in Figure 7. The observed results for Qwen1.5-MoE and Deepseek-MoE align closely with the theoretical predictions. In contrast, the data for Mixtral-8x7B shows a slight deviation, which is attributable to its violation of the independent distribution assumption. This analysis provides a clear explanation for the existence of fine-grained, element-wise similarity in expert weights, and reinforces the generalizability and explainability of our method.

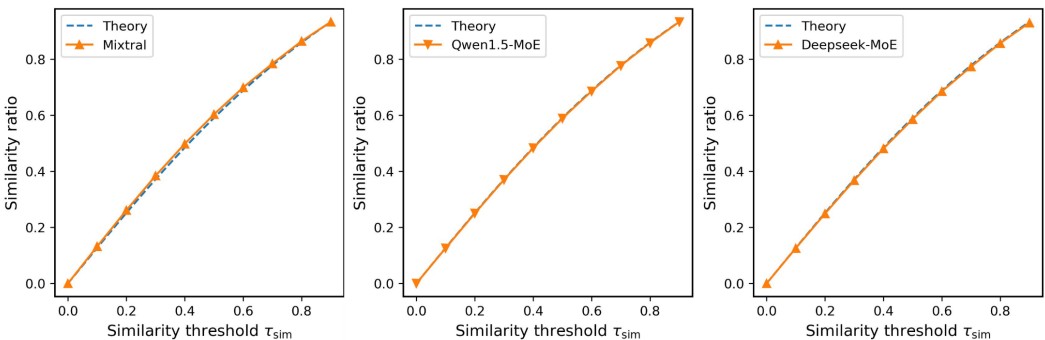

Figure 7: Relation between similarity ratio and similarity threshold $\tau_{\text{sim}}$.

### B.3 MORE ABLATION STUDY

#### B.3.1 COMBINING PUZZLEMOE WITH QUANTIZATION

We provide a detailed description of combining PuzzleMoE with quantization, as depicted in Figure 8. After the expert merging stage, the resultant floating-point weights are subjected to uniform quantization with a group size of 128. We employ a symmetric group quantization scheme, which obviates the need for a zero point. The final quantized values are stored alongside their corresponding sign and mask bits. The average bit width is subsequently determined by the following equation:

$$\text{Avg bitwidth} = (\underbrace{2}_{\text{Sign}} + \underbrace{1.58}_{\text{Compressed mask}} + \underbrace{3}_{\text{Quantize bitwidth}} + \underbrace{0.125}_{\text{Per-group scale}})/2 = 3.35$$

Each compressed mask belongs to the set $\{01,10,11\}$, as for each position, the merged weight either belongs to $W_i$, $W_j$, or both. Consequently, it requires only $\log_2 3 \approx 1.58$ bits for representation.

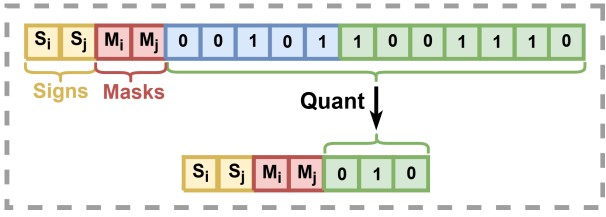

Figure 8: Combining PuzzleMoE with quantization.

We target an 80% compression ratio, as this represents a high level of compression without a substantial loss in model performance. Accordingly, we apply 3-bit quantization to the merged weights within the PuzzleMoE framework at 50% sparsity. To evaluate our approach, we compare the post-quantization performance of the expert modules using 3-bit AWQ (Lin et al., 2024) and NAEE/HC-SMoE with 25% sparsity + 4-bit AWQ. The results are shown in Table 10.

On Mixtral-8x7B, AWQ attains the highest average accuracy of 73.0 at 3.25 bits, while PuzzleMoE achieves a comparable 72.4 accuracy with a similar bitwidth of 3.35. Compared to NAEE+AWQ and HC-SMoE+AWQ, which both reach only 70.1 average accuracy, PuzzleMoE provides a clear gain of more than 2 points, indicating that expert merging via PuzzleMoE preserves quantization performance better than prior MoE compression schemes. On Deepseek-MoE, PuzzleMoE further improves both Wiki and average accuracy at a bitwidth of 3.35, whereas HC-SMoE+AWQ suffers a substantial drop to 53.3 average accuracy. Overall, these results suggest that PuzzleMoE can be seamlessly combined with low-bit quantization, delivering robust accuracy improvements over NAEE and HC-SMoE at only a negligible increase in effective bitwidth.

Table 10: Performance of combining PuzzleMoE with quantization.

| Model | Method | Avg bitwidth | Wiki | Avg Accuracy |
|-------|--------|-------------|------|-------------|
| Mixtral-8x7B | NAEE+AWQ | 3.19 | 5.10 | 70.1 |
| | AWQ | 3.25 | 4.41 | 73.0 |
| | HC-SMoE+AWQ | 3.30 | 5.39 | 70.1 |
| | PuzzleMoE | 3.35 | 4.50 | 72.4 |
| Deepseek-MoE | AWQ | 3.25 | 6.87 | 61.2 |
| | HC-SMoE+AWQ | 3.30 | 26.7 | 53.3 |
| | PuzzleMoE | 3.35 | 7.05 | 61.6 |

### B.3.2 IMPACT OF SALIENCY METRICS

To further investigate the effectiveness of the Wanda saliency metric, we evaluate several MoE-aware variants that incorporate router information to different extents: W ($|W|$), WG ($|W| \cdot \|gate\|_2$), WA ($|W| \cdot \|X\|_2$, identical to the original Wanda metric), and WAG ($|W| \cdot \|X \cdot gate\|_2$), where $W$ denotes the weight matrix, $X$ the activation, and $gate$ the router output.

As shown in Table 11, the activation-based variants (WA and WAG) achieve very similar overall performance and consistently match or slightly outperform the weight-only variants (W and WG). In contrast, adding gating scores on top of weight-only saliency (WG vs. W) brings almost no benefit. This indicates that the main performance gains stem from incorporating activation information into the saliency metric, which better preserves expert specialization, while additional router-gating signals provide no significant improvements.

Table 11: Performance of PuzzleMoE with different MoE-aware saliency metrics at 50% sparsity.

| Model | Method | Wiki | Avg |
|-------|--------|------|-----|
| Mixtral-8x7B | $|W|$ | 4.38 | 72.6 |
| | $|W| \cdot \|gate\|_2$ | 4.38 | 72.6 |
| | $|W| \cdot \|X\|_2$ | **4.36** | **72.6** |
| | $|W| \cdot \|X \cdot gate\|_2$ | 4.36 | 72.6 |
| DeepSeek-MoE | $|W|$ | 6.98 | 61.8 |
| | $|W| \cdot \|gate\|_2$ | 6.96 | 61.7 |
| | $|W| \cdot \|X\|_2$ | **6.88** | **62.1** |
| | $|W| \cdot \|X \cdot gate\|_2$ | 6.89 | 62.1 |

### B.4 CLARIFICATION ON BIT-PACKING SCHEME AND SPARSITY LEVEL

In this section we clarify the relationship between the proposed dual-mask expert merging algorithm and the bit-packing scheme, and discuss how PuzzleMoE can be extended beyond the 2-to-1 setting used in the main experiments.

PuzzleMoE consists of two conceptually separate parts:

1. **Dual-mask expert merging algorithm.** This is the core algorithmic contribution. It specifies how experts are selected and merged given two complementary binary masks, and is independent of the underlying numerical format or storage scheme.

2. **Bit-packing scheme that reuses `bfloat16` sign/exponent bits.** This is a systems optimization that stores merge masks by reusing the under-utilized sign/exponent bits of bfloat16 weights, making high-sparsity operation memory-efficient at scale.

In the main experiments we deliberately design the bit-packing scheme for 2-to-1 merging at 25% and 50% sparsity, in order to follow prior MoE merging works such as NAEE(Lu et al., 2024), HC-SMoE(Chen et al., 2025), and SubMoE(Li et al., 2025), which also focus on these sparsity regimes. Under these exact settings, PuzzleMoE incurs only about 2% accuracy degradation at 50% sparsity,

whereas prior methods typically suffer much larger drops (on average greater than 20%), making them difficult to deploy in practice.

**Extension to higher sparsity levels.** The current bit-packing design is indeed tailored to 2-to-1 merging. However, this does not constrain the dual-mask algorithm itself. If we store masks and signs explicitly without packing them into bfloat16 the same dual-mask merging algorithm can, in principle, support arbitrary compression ratios (e.g., 3-to-1, 4-to-1). In other words, the bit-packing scheme is introduced solely to reduce the memory footprint and make high-sparsity operation practical; it is not a limitation of the merging algorithm or of the achievable sparsity.

Table 12: Performance of PuzzleMoE and HC-SMoE on DeepSeek-MoE with 75% sparsity.

| Method | Wiki | ARC-C | ARC-E | Hella | PIQA | BoolQ | Wino | MMLU | Avg |
|--------|------|-------|-------|-------|------|-------|------|------|-----|
| PuzzleMoE | **14.04** | **33.4** | **65.3** | **54.1** | **72.2** | **62.2** | **66.4** | **24.9** | **50.4** |
| HC-SMoE | 8261 | 19.9 | 27.1 | 26.3 | 54.3 | 59.8 | 50.8 | 23.6 | 35.7 |

To empirically verify PuzzleMoE at higher sparsity levels, we perform additional experiments on DeepSeek-MoE where 4 experts are merged into 1 via hierarchical 2-to-1 merging, yielding a final sparsity of 75%. The results are summarized in Table 12. PuzzleMoE consistently outperforms the current SOTA HC-SMoE on almost all benchmarks and improves the average score from 35.7 to 50.4 (+14.7 points). These results suggest that PuzzleMoE remains effective even under substantially more aggressive compression. Designing a more flexible packed mask-storage scheme that supports sparsity levels beyond 50% while retaining the memory benefits of bit-packing is an interesting systems problem, and we leave such extensions to future work.

## B.5 TIME AND MEMORY COST OF PAIRWISE MERGING PROCESS

Table 13: Profiling of PuzzleMoE at 50% sparsity on Mixtral and DeepSeekMoE.

| Model | Others | Merge time (time/layer × #layers) | Forward time (time/layer × #layers) | Total | Memory |
|-------|--------|-----------------------------------|-------------------------------------|-------|--------|
| Mixtral-8x7B | 6s | 1.1s × 32 | 2.5s × 32 | 121s | 94.7GB |
| DeepSeek-MoE | 16s | 13.2s × 26 | 8.7s × 26 | 585s | 39.5GB |

We further detail the compression time and memory usage of the pairwise merging process in Table 13. In the case of Mixtral-8x7B, PuzzleMoE performs compression on two A100-80G GPUs without requiring parameter offloading. In contrast, NAEE and D2MoE require the storage of intermediate results during merging, leading to significant memory overhead. Consequently, these methods must offload model parameters, which introduces additional complexity and increases processing latency. This comparison shows that the pairwise merging stage in PuzzleMoE is both time-efficient and memory-efficient compared to the existing works, with overhead only marginally higher than loading and processing the original model weights.

## B.6 DETAILED RESULTS OF PUZZLEMOE WITH DIFFERENT SEEDS

The detailed accuracy results of PuzzleMoE with different random seeds on Mixtral-8x7B, Deepseek-MoE, Qwen1.5-MoE, and Qwen3-MoE are shown in Table 14, 15, 16, 17. Different seeds don't lead to a significant difference in accuracy performance, indicating the robustness of our method to different expert grouping choices.

Table 14: Zero-shot results of PuzzleMoE on Mixtral-8x7B under 25% and 50% sparsity with different seeds.

| Sparsity | Seed | Wiki | ARC-c | ARC-e | Hella | Piqa | BoolQ | Wino | MMLU | Avg |
|---|---|---|---|---|---|---|---|---|---|---|
| 0% | - | 3.84 | 56.7 | 84.1 | 64.9 | 82.4 | 85.4 | 77.2 | 67.9 | 74.1 |
| 25% | 1 | 4.09 | 55.4 | 83.4 | 64.1 | 82.2 | 85.5 | 75.3 | 66.6 | 73.2 |
| | 2 | 4.11 | 56.2 | 83.1 | 65.2 | 81.8 | 84.4 | 74.9 | 66.6 | 73.2 |
| | 3 | 4.11 | 55.1 | 82.8 | 64.0 | 82.2 | 85.6 | 75.4 | 67.1 | 73.2 |
| | 4 | 4.10 | 55.0 | 82.7 | 63.9 | 81.7 | 85.9 | 75.0 | 67.1 | 73.0 |
| | 5 | 4.10 | 54.8 | 83.1 | 64.0 | 81.7 | 85.4 | 75.4 | 66.6 | 73.0 |
| | 6 | 4.10 | 53.7 | 82.5 | 63.9 | 81.9 | 85.9 | 75.4 | 67.0 | 72.9 |
| | 7 | 4.09 | 56.3 | 83.4 | 64.1 | 82.5 | 85.4 | 74.8 | 66.7 | 73.3 |
| | 8 | 4.09 | 56.6 | 83.5 | 64.4 | 81.9 | 85.3 | 76.6 | 66.7 | 73.6 |
| | 9 | 4.09 | 55.2 | 83.5 | 64.2 | 82.2 | 85.3 | 75.9 | 66.6 | 73.3 |
| | 10 | 4.11 | 55.6 | 83.1 | 64.2 | 81.7 | 85.8 | 76.1 | 67.0 | 73.4 |
| | 11 | 4.10 | 55.0 | 82.8 | 64.2 | 82.3 | 85.3 | 75.0 | 66.4 | 73.0 |
| | 12 | 4.13 | 55.5 | 83.2 | 64.5 | 82.4 | 85.5 | 75.4 | 67.3 | 73.4 |
| | 13 | 4.10 | 54.6 | 83.3 | 64.1 | 82.1 | 85.8 | 75.5 | 66.5 | 73.1 |
| | 14 | 4.10 | 54.7 | 83.4 | 64.2 | 82.5 | 85.6 | 76.0 | 67.2 | 73.4 |
| | 15 | 4.09 | 56.5 | 83.6 | 64.3 | 81.8 | 84.1 | 76.5 | 66.9 | 73.4 |
| | 16 | 4.09 | 54.8 | 83.1 | 64.2 | 82.3 | 85.7 | 74.6 | 67.1 | 73.1 |
| | avg | 4.10 | 55.3 | 83.2 | 64.2 | 82.1 | 85.4 | 75.5 | 66.8 | 73.2 |
| | std | 0.01 | 0.8 | 0.3 | 0.3 | 0.3 | 0.5 | 0.6 | 0.3 | 0.2 |
| 50% | 1 | 4.36 | 53.1 | 82.5 | 63.3 | 82.1 | 84.6 | 76.4 | 65.8 | 72.5 |
| | 2 | 4.35 | 53.5 | 82.3 | 63.4 | 81.5 | 84.2 | 74.9 | 65.6 | 72.2 |
| | 3 | 4.35 | 53.3 | 82.2 | 63.4 | 81.6 | 86.5 | 75.9 | 66.0 | 72.7 |
| | 4 | 4.36 | 53.6 | 82.2 | 63.2 | 81.6 | 86.5 | 75.9 | 65.7 | 72.7 |
| | 5 | 4.36 | 54.3 | 82.3 | 63.2 | 81.5 | 85.1 | 75.5 | 65.5 | 72.5 |
| | 6 | 4.37 | 53.8 | 82.2 | 63.2 | 81.6 | 85.6 | 77.0 | 65.8 | 72.7 |
| | 7 | 4.37 | 54.0 | 82.7 | 63.4 | 82.1 | 85.8 | 75.9 | 65.8 | 72.8 |
| | 8 | 4.35 | 53.6 | 82.5 | 63.1 | 81.0 | 85.5 | 76.7 | 65.6 | 72.6 |
| | 9 | 4.35 | 54.4 | 82.5 | 63.2 | 81.8 | 85.2 | 75.8 | 65.4 | 72.6 |
| | 10 | 4.35 | 53.8 | 82.0 | 63.3 | 81.6 | 85.1 | 75.3 | 65.4 | 72.4 |
| | 11 | 4.35 | 53.6 | 82.3 | 63.4 | 81.6 | 83.6 | 75.5 | 65.9 | 72.3 |
| | 12 | 4.37 | 54.0 | 82.3 | 63.5 | 82.2 | 85.5 | 75.1 | 66.0 | 72.7 |
| | 13 | 4.36 | 53.9 | 82.5 | 63.3 | 81.8 | 85.7 | 76.2 | 65.1 | 72.6 |
| | 14 | 4.35 | 53.4 | 82.9 | 63.4 | 81.8 | 86.6 | 74.9 | 65.8 | 72.9 |
| | 15 | 4.37 | 53.7 | 82.8 | 63.3 | 81.1 | 84.2 | 75.9 | 66.2 | 72.5 |
| | 16 | 4.35 | 54.0 | 82.4 | 63.2 | 82.0 | 84.3 | 75.1 | 65.9 | 72.4 |
| | avg | 4.36 | 53.8 | 82.4 | 63.3 | 81.7 | 85.3 | 75.8 | 65.8 | 72.6 |
| | std | 0.01 | 0.3 | 0.2 | 0.1 | 0.3 | 0.9 | 0.6 | 0.3 | 0.2 |

Table 15: Zero-shot results of PuzzleMoE on Deepseek-MoE under 25% and 50% sparsity with different seeds.

| Sparsity | Seed | Wiki | ARC-c | ARC-e | Hella | Piqa | BoolQ | Wino | MMLU | Avg |
|---|---|---|---|---|---|---|---|---|---|---|
| 0% | - | 6.51 | 44.6 | 75.9 | 58.1 | 78.8 | 72.8 | 70.1 | 37.8 | 62.6 |
| 25% | 1 | 6.69 | 44.8 | 75.8 | 57.2 | 78.4 | 72.4 | 70.6 | 37.1 | 62.3 |
| | 2 | 6.67 | 43.9 | 75.9 | 57.3 | 78.2 | 70.7 | 70.5 | 35.4 | 61.7 |
| | 3 | 6.67 | 43.3 | 75.1 | 57.3 | 78.8 | 71.8 | 70.2 | 37.3 | 62.0 |
| | 4 | 6.68 | 44.4 | 75.7 | 57.5 | 78.9 | 71.0 | 70.6 | 37.2 | 62.2 |
| | 5 | 6.67 | 43.3 | 75.7 | 57.3 | 78.3 | 74.0 | 70.5 | 37.9 | 62.4 |
| | 6 | 6.67 | 43.8 | 75.6 | 57.1 | 78.7 | 72.9 | 71.0 | 38.0 | 62.4 |
| | 7 | 6.68 | 43.9 | 75.4 | 56.9 | 79.1 | 73.6 | 70.6 | 36.9 | 62.3 |
| | 8 | 6.70 | 44.1 | 75.6 | 56.8 | 78.8 | 75.1 | 70.7 | 36.2 | 62.5 |
| | 9 | 6.68 | 45.1 | 76.1 | 57.0 | 78.8 | 73.5 | 69.7 | 38.2 | 62.6 |
| | 10 | 6.68 | 44.2 | 75.4 | 57.3 | 78.4 | 73.7 | 69.9 | 37.0 | 62.3 |
| | 11 | 6.67 | 43.9 | 75.5 | 57.4 | 78.6 | 72.1 | 70.5 | 37.8 | 62.3 |
| | 12 | 6.66 | 44.0 | 76.2 | 57.2 | 78.3 | 72.5 | 70.9 | 36.2 | 62.2 |
| | 13 | 6.70 | 43.9 | 75.7 | 57.2 | 78.3 | 74.8 | 70.5 | 37.8 | 62.6 |
| | 14 | 6.70 | 43.4 | 76.4 | 57.4 | 79.2 | 73.8 | 70.6 | 36.2 | 62.4 |
| | 15 | 6.67 | 44.4 | 76.3 | 57.4 | 78.9 | 74.6 | 71.1 | 37.5 | 62.9 |
| | 16 | 6.67 | 43.9 | 75.4 | 57.4 | 78.9 | 73.3 | 70.9 | 39.2 | 62.7 |
| | avg | 6.68 | 44.0 | 75.7 | 57.2 | 78.7 | 73.1 | 70.6 | 37.2 | 62.4 |
| | std | 0.01 | 0.5 | 0.4 | 0.2 | 0.3 | 1.3 | 0.4 | 0.9 | 0.3 |
| 50% | 1 | 6.90 | 43.3 | 75.6 | 56.3 | 77.8 | 74.9 | 69.3 | 36.1 | 61.9 |
| | 2 | 6.89 | 42.8 | 75.9 | 56.3 | 78.3 | 73.4 | 70.5 | 35.5 | 61.8 |
| | 3 | 6.88 | 42.2 | 74.5 | 56.5 | 78.3 | 74.6 | 70.0 | 36.1 | 61.7 |
| | 4 | 6.87 | 43.1 | 75.8 | 56.7 | 78.5 | 75.0 | 69.5 | 37.2 | 62.3 |
| | 5 | 6.87 | 42.5 | 75.2 | 55.8 | 78.5 | 76.0 | 70.0 | 38.1 | 62.3 |
| | 6 | 6.88 | 41.9 | 74.6 | 56.6 | 78.4 | 74.7 | 71.2 | 37.2 | 62.1 |
| | 7 | 6.88 | 42.8 | 74.6 | 56.5 | 78.6 | 75.1 | 70.9 | 37.9 | 62.3 |
| | 8 | 6.89 | 43.5 | 75.1 | 56.5 | 78.2 | 73.7 | 69.9 | 36.7 | 61.9 |
| | 9 | 6.88 | 43.9 | 75.8 | 56.5 | 78.9 | 74.7 | 70.9 | 37.9 | 62.7 |
| | 10 | 6.89 | 43.3 | 74.8 | 56.5 | 78.7 | 74.9 | 71.0 | 36.9 | 62.3 |
| | 11 | 6.87 | 44.1 | 75.5 | 55.6 | 78.6 | 73.2 | 69.7 | 37.6 | 62.0 |
| | 12 | 6.85 | 43.1 | 74.8 | 56.4 | 77.8 | 74.3 | 69.6 | 36.2 | 61.7 |
| | 13 | 6.89 | 43.1 | 74.8 | 56.3 | 78.2 | 76.3 | 71.0 | 37.2 | 62.4 |
| | 14 | 6.89 | 41.5 | 75.9 | 55.7 | 78.2 | 72.4 | 70.6 | 34.3 | 61.2 |
| | 15 | 6.88 | 43.5 | 75.0 | 56.3 | 78.7 | 74.1 | 70.6 | 36.8 | 62.1 |
| | 16 | 6.87 | 42.7 | 74.7 | 56.5 | 78.8 | 75.2 | 69.9 | 38.6 | 62.3 |
| | avg | 6.88 | 43.0 | 75.2 | 56.3 | 78.4 | 74.5 | 70.3 | 36.9 | 62.1 |
| | std | 0.01 | 0.7 | 0.5 | 0.3 | 0.3 | 1.0 | 0.6 | 1.1 | 0.3 |

Table 16: Zero-shot results of PuzzleMoE on Qwen1.5-MoE under 25% and 50% sparsity with different seeds.

| Sparsity | Seed | Wiki | ARC-c | ARC-e | Hella | Piqa | BoolQ | Wino | MMLU | Avg |
|----------|------|------|-------|-------|-------|------|-------|------|------|-----|
| 0% | - | 7.22 | 41.0 | 73.2 | 58.0 | 80.0 | 79.5 | 68.9 | 61.0 | 65.9 |
| 25% | 1 | 7.38 | 40.9 | 73.4 | 57.2 | 79.8 | 79.5 | 69.1 | 60.5 | 65.8 |
| | 2 | 7.37 | 40.7 | 74.0 | 57.2 | 80.2 | 80.1 | 70.9 | 60.1 | 66.2 |
| | 3 | 7.39 | 41.6 | 72.4 | 57.2 | 78.9 | 78.9 | 69.0 | 60.4 | 65.5 |
| | 4 | 7.37 | 40.8 | 73.8 | 57.2 | 79.4 | 79.4 | 69.4 | 60.5 | 65.8 |
| | 5 | 7.38 | 40.1 | 72.8 | 57.3 | 79.8 | 78.6 | 70.9 | 60.2 | 65.7 |
| | 6 | 7.38 | 41.2 | 74.8 | 57.4 | 79.7 | 79.4 | 68.7 | 60.5 | 66.0 |
| | 7 | 7.37 | 41.4 | 73.5 | 57.7 | 79.8 | 78.8 | 70.6 | 60.3 | 66.0 |
| | 8 | 7.38 | 40.9 | 74.0 | 57.4 | 79.8 | 78.6 | 68.4 | 60.3 | 65.6 |
| | 9 | 7.37 | 40.0 | 73.2 | 57.2 | 80.3 | 78.8 | 69.4 | 60.2 | 65.6 |
| | 10 | 7.37 | 40.8 | 73.2 | 57.4 | 79.7 | 79.2 | 69.1 | 60.4 | 65.7 |
| | 11 | 7.37 | 40.8 | 73.7 | 57.5 | 80.1 | 79.4 | 69.9 | 60.3 | 66.0 |
| | 12 | 7.37 | 40.7 | 73.3 | 57.3 | 79.4 | 79.4 | 70.2 | 60.9 | 65.9 |
| | 13 | 7.37 | 40.7 | 72.6 | 57.3 | 79.4 | 79.0 | 69.3 | 60.8 | 65.6 |
| | 14 | 7.35 | 41.8 | 73.6 | 57.3 | 79.7 | 78.8 | 69.6 | 60.8 | 65.9 |
| | 15 | 7.39 | 40.6 | 73.3 | 57.2 | 79.4 | 79.7 | 68.6 | 60.2 | 65.6 |
| | 16 | 7.38 | 41.0 | 73.2 | 57.2 | 79.2 | 78.8 | 69.9 | 60.4 | 65.7 |
| | avg | 7.37 | 40.9 | 73.4 | 57.3 | 79.7 | 79.2 | 69.6 | 60.4 | 65.8 |
| | std | 0.01 | 0.5 | 0.6 | 0.1 | 0.4 | 0.4 | 0.8 | 0.2 | 0.2 |
| 50% | 1 | 7.55 | 39.9 | 72.7 | 56.3 | 80.4 | 80.2 | 69.0 | 59.9 | 65.5 |
| | 2 | 7.54 | 40.2 | 74.0 | 56.5 | 79.3 | 79.9 | 68.3 | 59.7 | 65.4 |
| | 3 | 7.54 | 40.8 | 73.1 | 56.5 | 79.3 | 79.9 | 69.6 | 60.0 | 65.6 |
| | 4 | 7.54 | 40.5 | 74.0 | 56.4 | 78.7 | 78.9 | 68.7 | 60.0 | 65.3 |
| | 5 | 7.55 | 41.3 | 73.2 | 56.6 | 79.4 | 76.2 | 69.3 | 59.7 | 65.1 |
| | 6 | 7.54 | 40.7 | 74.3 | 56.3 | 79.3 | 79.6 | 69.1 | 59.8 | 65.6 |
| | 7 | 7.56 | 42.4 | 75.4 | 56.7 | 78.9 | 77.6 | 69.5 | 60.3 | 65.8 |
| | 8 | 7.54 | 40.6 | 74.1 | 56.5 | 79.6 | 78.4 | 69.8 | 60.0 | 65.6 |
| | 9 | 7.55 | 39.7 | 72.2 | 56.7 | 79.7 | 77.8 | 69.9 | 59.0 | 65.0 |
| | 10 | 7.56 | 40.7 | 72.9 | 56.6 | 79.5 | 78.1 | 70.1 | 60.5 | 65.5 |
| | 11 | 7.54 | 39.3 | 72.7 | 56.6 | 79.9 | 78.9 | 69.1 | 59.7 | 65.2 |
| | 12 | 7.54 | 39.9 | 73.2 | 56.5 | 78.9 | 78.0 | 69.7 | 60.5 | 65.2 |
| | 13 | 7.54 | 41.7 | 73.4 | 56.8 | 79.1 | 79.1 | 70.0 | 60.2 | 65.8 |
| | 14 | 7.54 | 41.6 | 74.2 | 56.7 | 79.3 | 77.3 | 68.4 | 60.0 | 65.4 |
| | 15 | 7.55 | 40.5 | 73.6 | 56.4 | 79.2 | 79.5 | 69.7 | 60.4 | 65.6 |
| | 16 | 7.55 | 40.9 | 73.4 | 56.6 | 79.4 | 78.7 | 70.4 | 59.5 | 65.6 |
| | avg | 7.55 | 40.7 | 73.5 | 56.5 | 79.4 | 78.6 | 69.4 | 60.0 | 65.4 |
| | std | 0.01 | 0.8 | 0.8 | 0.1 | 0.4 | 1.1 | 0.6 | 0.4 | 0.2 |

Table 17: Zero-shot results of PuzzleMoE on Qwen3-MoE under 25% and 50% sparsity with different seeds.

| Sparsity | Seed | Wiki | ARC-c | ARC-e | Hella | Piqa | BoolQ | Wino | MMLU | Avg |
|---|---|---|---|---|---|---|---|---|---|---|
| 0% | - | 8.70 | 52.7 | 79.3 | 59.5 | 79.6 | 88.7 | 70.4 | 77.8 | 72.6 |
| 25% | 1 | 9.17 | 49.3 | 76.5 | 57.9 | 79.2 | 87.6 | 71.0 | 76.6 | 71.2 |
| | 2 | 9.03 | 50.8 | 77.8 | 58.5 | 78.7 | 88.2 | 70.4 | 77.1 | 71.6 |
| | 3 | 9.07 | 51.8 | 79.3 | 58.4 | 79.2 | 87.9 | 68.8 | 76.1 | 71.6 |
| | 4 | 9.07 | 51.1 | 78.5 | 58.2 | 79.0 | 88.3 | 71.0 | 76.8 | 71.8 |
| | 5 | 9.12 | 54.3 | 79.2 | 58.5 | 79.8 | 87.8 | 70.2 | 76.4 | 72.3 |
| | 6 | 9.14 | 50.7 | 78.4 | 58.2 | 78.8 | 88.5 | 70.9 | 77.1 | 71.8 |
| | 7 | 9.17 | 51.9 | 79.3 | 58.4 | 79.3 | 88.3 | 69.5 | 76.3 | 71.9 |
| | 8 | 9.05 | 53.1 | 79.5 | 58.4 | 79.1 | 88.4 | 70.8 | 76.8 | 72.3 |
| | 9 | 9.09 | 51.0 | 78.9 | 58.1 | 79.1 | 88.3 | 70.4 | 76.5 | 71.8 |
| | 10 | 9.11 | 52.7 | 81.0 | 58.5 | 79.4 | 88.6 | 70.3 | 76.2 | 72.4 |
| | 11 | 9.04 | 51.5 | 78.8 | 58.2 | 79.3 | 87.7 | 69.9 | 76.9 | 71.8 |
| | 12 | 9.07 | 50.3 | 78.8 | 58.4 | 79.8 | 88.1 | 71.2 | 76.6 | 71.9 |
| | 13 | 8.97 | 51.5 | 79.3 | 58.1 | 80.0 | 88.8 | 69.8 | 76.5 | 72.0 |
| | 14 | 9.11 | 51.0 | 79.2 | 58.2 | 79.3 | 88.7 | 70.8 | 75.9 | 71.9 |
| | 15 | 9.11 | 50.7 | 78.2 | 58.2 | 78.6 | 88.0 | 70.9 | 76.8 | 71.6 |
| | 16 | 9.03 | 53.3 | 79.6 | 58.5 | 79.4 | 88.6 | 70.2 | 76.4 | 72.3 |
| | avg | 9.08 | 51.6 | 78.9 | 58.3 | 79.3 | 88.2 | 70.4 | 76.6 | 71.9 |
| | std | 0.05 | 1.3 | 1.0 | 0.2 | 0.4 | 0.4 | 0.6 | 0.3 | 0.3 |
| 50% | 1 | 9.46 | 50.1 | 77.7 | 56.9 | 78.7 | 87.0 | 70.4 | 74.7 | 70.8 |
| | 2 | 9.57 | 54.8 | 79.4 | 57.0 | 78.7 | 87.8 | 69.5 | 75.3 | 71.8 |
| | 3 | 9.54 | 50.9 | 77.8 | 56.8 | 78.9 | 87.4 | 69.7 | 75.1 | 70.9 |
| | 4 | 9.46 | 49.5 | 77.0 | 57.1 | 79.1 | 87.9 | 71.4 | 75.5 | 71.1 |
| | 5 | 9.59 | 52.3 | 79.1 | 56.8 | 79.1 | 88.0 | 70.8 | 75.5 | 71.7 |
| | 6 | 9.53 | 50.6 | 78.0 | 57.3 | 78.7 | 88.0 | 70.2 | 74.9 | 71.1 |
| | 7 | 9.45 | 50.2 | 79.4 | 57.3 | 78.2 | 88.1 | 70.6 | 74.9 | 71.2 |
| | 8 | 9.51 | 50.4 | 78.3 | 56.9 | 79.1 | 88.4 | 70.6 | 75.2 | 71.3 |
| | 9 | 9.53 | 51.6 | 78.0 | 57.3 | 79.0 | 88.1 | 70.0 | 74.7 | 71.2 |
| | 10 | 9.47 | 49.5 | 79.6 | 57.0 | 78.9 | 88.0 | 68.8 | 74.4 | 70.9 |
| | 11 | 9.51 | 50.9 | 79.0 | 57.1 | 78.9 | 87.9 | 69.8 | 75.2 | 71.3 |
| | 12 | 9.44 | 50.7 | 79.6 | 57.3 | 78.5 | 87.9 | 70.6 | 75.6 | 71.5 |
| | 13 | 9.46 | 50.5 | 78.5 | 56.9 | 78.8 | 88.1 | 69.9 | 74.9 | 71.1 |
| | 14 | 9.43 | 52.4 | 79.8 | 57.4 | 79.9 | 88.4 | 69.5 | 75.6 | 71.9 |
| | 15 | 9.58 | 47.4 | 76.6 | 57.3 | 78.4 | 87.7 | 68.9 | 74.7 | 70.1 |
| | 16 | 9.53 | 53.9 | 78.5 | 57.1 | 79.2 | 88.8 | 70.1 | 75.2 | 71.8 |
| | avg | 9.50 | 51.0 | 78.5 | 57.1 | 78.9 | 88.0 | 70.1 | 75.1 | 71.2 |
| | std | 0.05 | 1.8 | 1.0 | 0.2 | 0.4 | 0.4 | 0.7 | 0.4 | 0.4 |