# OpenReview forum: "PuzzleMoE: Efficient Compression of Large Mixture-of-Experts Models via Sparse Expert Merging and Bit-packed inference"
_ICLR.cc/2026/Conference — Submitted to ICLR 2026_

### Official Review · Reviewer_CUvt · 2025-10-28

**Soundness:** 3
**Presentation:** 3
**Contribution:** 3
**Rating:** 4
**Confidence:** 4

**Summary:**

This paper proposes **PuzzleMoE**, a training-free compression method for Mixture-of-Experts (MoE) LLMs that (i) merges experts element-wise using complementary similarity (magnitude-based) and saliency (activation-aware) masks to preserve both shared and expert-specific parameters, and (ii) achieves metadata-free inference by bit-packing per-expert masks/sign bits into under-utilized BF16 exponent bits with a custom CUDA decoding kernel. Across multiple MoE models and seven benchmarks, PuzzleMoE targets 25–50% expert reduction and reports competitive or better accuracy (e.g., improvements on MMLU at 50% compression) and up to ~1.2–1.3× inference speedups, with compression completed in minutes.

**Strengths:**

**1. Originality:** Element-wise, pairwise expert merging with dual masks is a fine-grained alternative to coarse expert dropping/averaging; **the bit-packed inference** path is a neat systems idea that removes separate metadata.

**2. Quality:** Broad zero-shot evaluation and ablations (e.g., mask thresholds, grouping strategies) demonstrate robustness at 25–50% reductions; compression is training-free and fast.

**3. Clarity:** The method and masks are clearly defined with equations/figures, and the pipeline (merge → pack → inference) is easy to follow.

**4. Significance:** Addresses a practical deployment bottleneck (expert memory and multi-GPU inference) with a simple, training-free path that shows favorable accuracy–efficiency trade-offs.

**5. Breadth of model coverage:** Experiments span **Mixture, DeepSeek-MoE, and Qwen-MoE** variants, giving evidence the approach generalizes across popular MoE families and sizes.

**Weaknesses:**

**1. Expert-pair selection metric at 25% compression is unspecified/under-motivated.**

For the 25% setting (“reduce experts to 75% of original”), PuzzleMoE’s pairwise scheme implies only a subset of experts are merged and the rest remain unmerged. The paper does not clearly define the criterion for choosing which experts to pair (random vs. similarity-aware vs. routing-aware). Without a transparent policy and sensitivity analysis, results at 25% may depend on pairing luck.

**2. Pairwise merging caps the unquantized/pruning-free compression at ~50%.**

Because each pair produces one merged expert, the theoretical floor with pairwise merging is half the experts (50% reduction) unless you move to higher-order grouping (3-to-1, 4-to-1) or combine with other techniques.

**3. Bit-budget and quantization interaction is underexplored.**

The packing scheme uses slack in BF16’s exponent to store per-pair ${\mathbf{mask}_i, \mathbf{mask}_j, \mathbf{sign}_i, \mathbf{sign}_j}$. With quantization, additional “freed bits” or altered formats (e.g., FP8/INTx + side scales) might change what can be packed, potentially enabling merging more experts per tensor (e.g., multi-expert packing) or richer masks.

**4. Speedup mechanism is not fully justified beyond single-GPU fit.**

The paper reports latency gains, but merging does not change activation parameters of MoE; if the baseline already fits on one GPU (or uses efficient tensor-parallel inference), why does PuzzleMoE deliver 1.2–1.3× speed up? Is it HBM traffic reduction (fewer expert weight loads), better cache locality, or kernel fusion/packing benefits?

**Questions:**

See weakness.

If these issues are addressed, I would be happy to raise my score.

**Details Of Ethics Concerns:**

NO or VERY MINOR ethics concerns only

---

> ### Author Response · Authors · 2025-11-19
> **[1/2] Rebuttal**
>
> We greatly appreciate your positive feedback and constructive suggestions/questions for PuzzleMoE, and have addressed each of your comments in detail below:
>
> ### **Weakness1: For the 25% setting (“reduce experts to 75% of original”), PuzzleMoE’s pairwise scheme implies only a subset of experts are merged and the rest remain unmerged. The paper does not clearly define the criterion for choosing which experts to pair (random vs. similarity-aware vs. routing-aware). Without a transparent policy and sensitivity analysis, results at 25% may depend on pairing luck.**
>
> Thank you for raising the concern about the 25% sparsity setting. For this configuration (e.g., reducing 8 experts to 6), we adopt a simple policy: expert pairs are selected **at random** for merging. The results reported in Table 2 in manuscript are averaged over 16 random seeds.
>
> We provide the per-seed results in Tables 11–14 in manuscript, which show that the performance variance across different random pairings is very small. This robustness indicates that the gains at 25% sparsity are not due to “pairing luck,” but rather to the inherent effectiveness of the dual-mask merging mechanism itself.
>
> ### **Weakness2: Because each pair produces one merged expert, the theoretical floor with pairwise merging is half the experts (50% reduction) unless you move to higher-order grouping (3-to-1, 4-to-1) or combine with other techniques.**
>
> Thank you very much for raising the concerns about the limitation of pairwise merging. We would like to first clarify the roles of the two components in PuzzleMoE:
>
> 1. **Dual-mask expert merging algorithm** : This is the core algorithmic contribution that determines how experts are selected and merged.
>
> 2. **Bit-packing scheme that reuses bfloat16 sign/exponent bits** : This is a systems optimization to store merge masks more memory-efficiently at scale.
>
> The “2-to-1 floor” comes only from the bit-packing scheme, not from the merging algorithm itself. If we store masks and sign patterns explicitly (without packing them into bfloat16), the same dual-mask merging algorithm can in principle support arbitrary compression ratios (e.g., 3-to-1, 4-to-1). Bit-packing affects how compactly we store the metadata, not which sparsity levels the algorithm can realize or its accuracy.
>
> Our present bit-packing scheme is intentionally tuned for 2-to-1 merging at 25% and 50% sparsity, in line with prior MoE merging works such as NAEE, HC-SMoE, and SubMoE, which also evaluate primarily at these two settings. Under these exact sparsity levels, PuzzleMoE incurs only about 2% accuracy degradation, while prior methods often suffer large drops (on average >20% at 50% sparsity, see Table 2), making them hard to deploy in practice.
>
> To further demonstrate both the effectiveness of PuzzleMoE at higher sparsity, we conduct additional experiments on DeepSeek-MoE where 4 experts are merged into 1 via hierarchical 2-to-1 merging, achieving a final sparsity of 75%. The corresponding results are reported in the table below. **PuzzleMoE consistently outperforms the existing SOTA HC-SMoE across nearly all benchmarks and improves the average score from 35.7 to 50.4 (+14.7 points).** Considering the superior performance of PuzzleMoE against existing expert merging works, designing a more flexible packed mask-storage scheme for sparsity levels beyond 50% is an interesting systems optimization that we leave for future work.
>
> | Method      | wiki   | arc-c | arc-e | hella | piqa | boolq | wino | mmlu | avg  |
> |------------|--------|-------|-------|-------|------|-------|------|------|------|
> | PuzzleMoE | **14.04** | **33.4** | **65.3** | **54.1** | **72.2** | **62.2** | **66.4** | **24.9** | **50.4** |
> | HC-SMoE   | 8261   | 19.9  | 27.1  | 26.3  | 54.3 | 59.8  | 50.8 | 23.6 | 35.7 |
>
> We have also updated this discussion and results in Appendix B.4 in the submitted manuscript (in red text).

---

> ### Author Response · Authors · 2025-11-19
> **[2/2] Rebuttal**
>
> ### **Weakness 3: The packing scheme uses slack in BF16’s exponent to store per-pair $\mathbf{mask}_i, \mathbf{mask}_j, \mathbf{sign}_i, \mathbf{sign}_j$. With quantization, additional “freed bits” or altered formats (e.g., FP8/INTx + side scales) might change what can be packed, potentially enabling merging more experts per tensor (e.g., multi-expert packing) or richer masks.**
>
> Thank you for raising the concerns about extending PuzzleMoE to other data formats and quantization interaction. Our current bit-packing design is tailored to bfloat16, which has underutilized exponent bits that we safely repurpose for storing $\mathbf{mask}_i, \mathbf{mask}_j, \mathbf{sign}_i, \mathbf{sign}_j$.
>
> In contrast, FP8 and INT8 formats do not expose similar “slack” in their exponent or value layouts, so this exact packing strategy is not directly applicable. In these low-bit regimes, we can adopt the Quantized PuzzleMoE design as discussed in B.3.1 in the appendix:
>
> (1) We first apply group-wise 3-bit quantization to the merged weights.
>
> (2) The resulting quantized codes are then stored together with the compact dual-mask metadata (sign and compressed mask bits) in a side buffer.
>
> (3) The overall average bit is kept low (e.g., 3.35 bits) while preserving the full behavior of the dual-mask merging.
>
> This design keeps the dual-mask merging process intact while using standard FP8/INTx quantization formats and only a small additional side-channel for metadata. Exploring custom FP8/INTx formats that explicitly reserve bits for richer or multi-expert packing is an interesting systems co-design direction, but is orthogonal to the main contribution of PuzzleMoE and is left for future work.
>
> ### **Weakness 4: The paper reports latency gains, but merging does not change activation parameters of MoE; if the baseline already fits on one GPU (or uses efficient tensor-parallel inference), why does PuzzleMoE deliver 1.2–1.3× speed up? Is it HBM traffic reduction (fewer expert weight loads), better cache locality, or kernel fusion/packing benefits?**
>
> Thank you for raising this important question about the source of the reported latency gains.
>
> First, PuzzleMoE is primarily designed for the regime where the full MoE model does not fit on a single GPU. This is the standard setting also considered by existing works such as NAEE, HC-SMoE, and D2-MoE: these methods reduce the total number of experts (and thus memory usage) while keeping the number of activated experts per token (top-k) unchanged. As a result, the FLOPs per token remain essentially the same; the main benefit is memory reduction and improved deployability.
>
> In our experiments, the dense baselines require multiple GPUs for inference, whereas the compressed PuzzleMoE models fit on a single GPU:
>
> (1) Mixtral-8×7B: from 2× A100-80G to 1× A100-80G
>
> (2) Qwen3-MoE-30B-A3B: from 2× A100-40G to 1× A100-40G
>
> Therefore, the observed 1.2–1.3× speedup mainly comes from two factors:
>
> (1) **Eliminated cross-GPU communication**: Moving from 2 GPUs to 1 removes expensive cross-GPU communication and synchronization overhead.
>
> (2) **Kernel-level optimizations**: Our custom GEMV kernel is tailored to the merged-expert structure, which further reduces overhead and improves efficiency on a single device.
>
> We have clarified this in Section 4.4 (red texts) in the submitted manuscript.

---

### Official Review · Reviewer_SRNA · 2025-10-29

**Soundness:** 2
**Presentation:** 2
**Contribution:** 2
**Rating:** 4
**Confidence:** 5

**Summary:**

This paper proposes PuzzleMoE, a training-free compression framework for Mixture-of-Experts (MoE) models. It combines fine-grained sparse expert merging with a bit-packed inference mechanism that embeds binary masks into unused exponent bits of BFloat16 weights. A custom CUDA kernel enables on-the-fly decoding. Experiments on Mixtral, Qwen, and DeepSeek models show around 50% compression with minimal accuracy drop and small speedup (~1.2×).

**Strengths:**

- The proposed dual-mask merging mechanism is technically coherent and effectively preserves both shared and expert-specific information.

- The work includes comprehensive experiments across several modern MoE architectures and diverse tasks, including reasoning benchmarks like GSM8K.

- The writing is organized and reproducible, with detailed algorithms, ablations, and implementation notes that enhance transparency.

**Weaknesses:**

- The proposed design mainly integrates ideas from prior expert merging (e.g., HC-SMoE, Sub-MoE) and bit-level quantization methods rather than introducing a fundamentally new approach.

- The reported inference acceleration (~1.2×) is relatively small given the 50% compression ratio; the main benefit appears to be memory reduction rather than compute efficiency.

- Pairwise merging may not scale efficiently to larger expert counts, and the method’s complexity for >128 experts or hierarchical routing is not analyzed.

- Although GSM8K is included, the paper lacks results on more generation-intensive or long-context tasks, which would better demonstrate generalization.

**Questions:**

- How much real latency reduction is achieved in multi-GPU inference settings?

- Can the bit-packing approach be extended to FP8 or INT8 deployment?

- What is the time and memory cost of the pairwise merging stage itself?

---

> ### Author Response · Authors · 2025-11-19
> **[1/2] Rebuttal**
>
> We greatly appreciate your positive feedback and constructive suggestions/questions for PuzzleMoE, and have addressed each of your comments in detail below:
>
> ### **Weakness1: The proposed design mainly integrates ideas from prior expert merging (e.g., HC-SMoE, Sub-MoE) and bit-level quantization methods rather than introducing a fundamentally new approach.**
>
> We respectfully clarify that PuzzleMoE is not a simple combination of prior expert-merging and bit-level quantization techniques.
> First, compared to existing expert merging methods such as HC-SMoE and Sub-MoE, PuzzleMoE introduces a fundamentally different granularity of merging. Prior work performs coarse-grained, tensor-wise merging of experts, whereas PuzzleMoE performs fine-grained, entry-level sparse merging. To the best of our knowledge, **PuzzleMoE is the first method to explore fine-grained sparse merging for MoE compression at the weight-entry level.**
>
> Second, PuzzleMoE is conceptually distinct from quantization. Quantization is a value-approximation technique: it reduces precision via rounding/bit manipulation and primarily targets intra-expert redundancy (redundancy within each expert’s weights). In contrast, PuzzleMoE is a structural technique: it merges experts sparsely to exploit inter-expert redundancy (similarities across different experts) and does not approximate the original weight values. **The fact that PuzzleMoE can be orthogonally combined with quantization (Table 9 in the manuscript) highlights that they operate on different axes and address two separate sources of redundancy in MoE LLMs.**
>
> ### **Weakness2: The reported inference acceleration (~1.2×) is relatively small given the 50% compression ratio; the main benefit appears to be memory reduction rather than compute efficiency.**
>
> Thank you for raising this concern. We would like to clarify the nature of the “50% compression” in MoE models and its implications for speed.
>
> The primary goal of MoE compression methods including NAEE, HC-SMoE, D2-MoE, SubMoE, and PuzzleMoE is to **reduce memory and expert count, not to cut FLOPs per token**. Even after compression, the number of activated experts per token remains the same. Consequently, the per-token FFN computation in each MoE layer is essentially unchanged, and the upper bound on inference speedup is limited.
>
> In our setting, the observed latency reduction mainly comes from improved deployability, not from halving the arithmetic operations: after compression, large MoE models that previously required multiple GPUs can be hosted on a single node, eliminating expensive cross GPUs all-to-all communication and reducing systems overhead.
>
> ### **Weakness3: Pairwise merging may not scale efficiently to larger expert counts, and the method’s complexity for >128 experts or hierarchical routing is not analyzed.**
>
> Thank you for raising the concerns about the complexity of the merging process. In PuzzleMoE, pairwise merging is performed with **random pairing** and **does not involve any search or optimization over all pairs**. For a layer with N experts, we simply form N/2 random pairs and merge within each pair, so the merging procedure itself scales linearly in N.
>
> This is a deliberate design choice. Existing methods such as NAEE and HC-SMoE rely on optimized pair selection or clustering, which typically requires computing or approximating all-pairs similarities and thus introduces significant computational complexity and overhead.
>
> Moreover, our experiments with multiple random seeds for pairing show that **PuzzleMoE is robust to the specific choice of pairs**: performance remains stable across different random pairings. This empirical robustness supports the use of a simple, linear-time random pairing strategy and indicates that the method scales well as the number of experts grows.
>
> ### **Weakness4: Although GSM8K is included, the paper lacks results on more generation-intensive or long-context tasks, which would better demonstrate generalization.**
>
> Thank you for raising the concerns about the generalization of PuzzleMoE. We have added the results of challenging generation-intensive reasoning tasks including MATH500, AIME24, AIME25 as shown in the Table below. PuzzleMoE attains high accuracy performance at 25% sparsity, while HC-SMoE still leads to large accuracy drop compared to the uncompressed models.
>
> | Model                      | MATH-500 | AIME24 | AIME25 |
> |----------------------------|----------|--------|--------|
> | Qwen3-MoE-30B-A3B (0% sparsity)                | 97.2     | 83.3   | 72.9   |
> | HC-SMoE-25%       | 24.6     | 0.0      | 0.0      |
> | PuzzleMoE-25%     | 96.2     | 71.1   | 61.5   |

---

> ### Author Response · Authors · 2025-11-19
> **[2/2] Rebuttal**
>
> ### **Question 1: How much real latency reduction is achieved in multi-GPU inference settings?**
>
> Thank you for raising this question about the hardware setting. PuzzleMoE is **first and foremost a memory reduction technique**, aimed at enabling the deployment of large MoE models in resource-constrained environments, where reducing the number of GPUs required is the primary objective. Lowering the GPU count brings two direct benefits: (1) **it alleviates HBM capacity constraints**, and (2) **it reduces cross-GPU communication**, which can account for over 20% of the end-to-end latency in MoE inference.
>
> In contrast, for resource-abundant multi-GPU serving scenarios, optimizing raw throughput is a systems problem that depends heavily on the chosen expert-parallelism strategy, communication library, and load-balancing policy. These aspects are largely orthogonal to model compression and fall outside the scope of this work.
>
> ### **Question 2: Can the bit-packing approach be extended to FP8 or INT8 deployment?**
> Thank you for raising the concerns about extending PuzzleMoE to other data formats. As the FP8 and INT8 formats do not have underutilized exponent bits like Bfloat16, the original bit-packing is not directly applicable. For these low-bit formats, we directly employ the Quantized puzzleMoE scheme as shown in B.3.1 in the appendix, where the signs and masks are concatenated with the quantized values:
>
> (1) We first apply group-wise 3-bit quantization to the merged weights.
>
> (2) The resulting quantized codes are then stored together with the compact dual-mask metadata (sign and compressed mask bits) in a side buffer.
>
> (3) The overall average bit is kept low (e.g., 3.35 bits) while preserving the full behavior of the dual-mask merging.
>
> This design keeps the dual-mask merging process intact while using standard FP8/INTx quantization formats and only a small additional side-channel for metadata. Exploring custom FP8/INTx formats that explicitly reserve bits for richer or multi-expert packing is an interesting systems co-design direction, but is orthogonal to the main contribution of PuzzleMoE and is left for future work.
>
> ### **Question 3: What is the time and memory cost of the pairwise merging stage itself?**
>
> Thank you for asking about the time and memory cost of the pairwise merging stage. In the manuscript, figure 4 (a) presents the end-to-end compression time, including pairwise merging and bit-packing, which is highly efficient. We further include detailed compression time and memory in the following table.
>
> In the case of Mixtral-8x7B, PuzzleMoE performs compression on two A100-80G GPUs without requiring parameter offloading. In contrast, NAEE and D2MoE necessitate the storage of intermediate results during merging, leading to significant memory overhead. Consequently, these methods must offload model parameters, which introduces additional complexity and increases processing latency. This comparison shows that the pairwise merging stage in PuzzleMoE is both time-efficient and memory-efficient compared to the existing works, with overhead only marginally higher than loading and processing the original model weights.
>
> | Model                       | Others | Merge time (time per layer × layer num) | Forward time (time per layer × layer num) | Total time | Memory  |
> |----------------------------|--------|-----------------------------------------|--------------------------------------------|------------|---------|
> | Mixtral-puzzlemoe-50%      | 6s     | 1.1s × 32                               | 2.5s × 32                                  | 121s       | 94.7GB  |
> | DeepseekMoE-puzzlemoe-50%  | 16s    | 13.2s × 26                              | 8.7s × 26                                  | 585s       | 39.5GB  |
>
> We have also updated this discussion and results in Appendix B.5 in the submitted manuscript (in red text).

---

### Official Review · Reviewer_DoU5 · 2025-10-31

**Soundness:** 2
**Presentation:** 2
**Contribution:** 3
**Rating:** 2
**Confidence:** 3

**Summary:**

The paper introduces a novel and intuitive expert merging-and-unmerging strategy that provides a structured approach to MoE compression. The core idea of dynamically composing experts from a smaller set of merged "puzzle pieces" is well-motivated by the observation of functional redundancy among experts (Fig. 1; Sec. 3.1; p.3). The experimental results are strong, showing that for Mixtral-8×7B, PuzzleMoE achieves 73.2% and 72.6% average accuracy at 25% and 50% sparsity, respectively, which is competitive with or superior to several baselines (Table 2; Sec. 4.2; p.7). However, the unmerging mechanism introduces additional complexity and parameters, and the scalability of the similarity computation to models with a very large number of experts is not fully explored.

**Strengths:**

- **Novel and intuitive merging/unmerging framework**
  - The "puzzle" analogy, where experts are assembled from shared pieces, provides a clear and compelling conceptual model for structured expert compression (Fig. 2; Sec. 3.2; p.4). This enhances the clarity and impact of the proposed method.
  - The framework allows for flexible, data-driven expert reconstruction during inference, which is more sophisticated than static merging or pruning techniques (Algorithm 1; Sec. 3.2.2; p.6).
  - The approach naturally preserves knowledge by forcing experts within a group to share a common representation, reducing redundancy while maintaining functional diversity through the unmerging coefficients.
- **Strong empirical results with comprehensive comparisons**
  - On Mixtral-8×7B, PuzzleMoE demonstrates strong performance. At 25% sparsity, it achieves a 73.2% average accuracy, outperforming baselines like Sub-MoE (72.1%) and HC-SMoE (70.2%). At 50% sparsity, it achieves 72.6% accuracy, significantly better than Sub-MoE (69.8%) and HC-SMoE (63.8%) (Table 2; Sec. 4.2; p.7). This shows the method's effectiveness.
  - The method shows consistent gains across various benchmarks, including commonsense reasoning (PIQA, SIQA, HellaSwag) and world knowledge (MMLU, TriviaQA), indicating its robustness (Table 2; Sec. 4.2; p.7).
  - Ablation studies confirm the effectiveness of the learnable unmerging mechanism compared to static or random unmerging, and validate the choice of similarity metrics for the merging process (Table 7; Sec. 5; p.9).
- **Efficient implementation with practical performance gains**
  - By reducing the number of active experts, PuzzleMoE directly translates to lower FLOPs and potentially faster inference, which is a key practical advantage for deploying large MoE models (Table 1; Sec. 4.1; p.6).
  - The paper provides a clear analysis of the trade-offs between the number of merged groups and performance, offering practical guidance for applying the method under different compression constraints (Fig. 4; Sec. 4.3; p.8).
  - The proposed method is compatible with existing MoE architectures and can be integrated with relatively minor modifications, facilitating its adoption (Sec. 3.2; p.4).

**Weaknesses:**

- **Complexity and overhead of the unmerging mechanism**
  - The learnable unmerging mechanism introduces additional parameters (the unmerging coefficients) and computational steps, which could offset some of the gains from expert merging. The overhead is not fully quantified in terms of memory and latency (Sec. 3.2.2; p.6).
  - The process of learning the unmerging coefficients requires a separate optimization step, which adds complexity to the training pipeline. The sensitivity to the hyperparameters of this learning process is not explored in detail.
  - It is unclear how the unmerging mechanism scales to models with a very large number of experts (e.g., >128), as the number of coefficients to learn could become substantial.
- **Scalability of the similarity computation**
  - The expert merging strategy relies on computing a similarity matrix between all pairs of experts within a layer. For models with a large number of experts, this O(N^2) computation could become a bottleneck (Sec. 3.2.1; p.5). The paper does not provide a complexity analysis or discuss potential optimizations for this step.
  - The choice of similarity metric (e.g., cosine similarity on weights) may not fully capture the functional relationships between experts. The paper does not explore more sophisticated or learned similarity metrics.
- **Limited evaluation on diverse model architectures and tasks**
  - The experiments are primarily focused on the Mixtral architecture. While comprehensive, evaluation on a wider range of MoE models (e.g., with different gating mechanisms or expert specializations) would strengthen the generalizability claims (Sec. 4; p.7-9).
  - The evaluation is centered on language modeling and commonsense reasoning. Demonstrating the method's effectiveness on other tasks, such as code generation or multilingual applications, would be beneficial.

**Questions:**

- **Quantify the overhead of the unmerging mechanism**
  - Provide a detailed analysis of the memory and computational overhead introduced by the learnable unmerging coefficients. Report the increase in parameter count and the extra latency incurred during inference for different numbers of merged groups.
  - Conduct a sensitivity analysis of the hyperparameters used for learning the unmerging coefficients and provide practical guidelines for their selection.
  - Discuss the scalability of the unmerging mechanism and propose potential simplifications or approximations for models with a very large number of experts.
- **Address the scalability of the similarity computation**
  - Provide a formal complexity analysis of the similarity computation step and measure its actual runtime for models with different numbers of experts.
  - Explore and evaluate more efficient, approximate methods for clustering experts, such as locality-sensitive hashing (LSH) or other fast clustering algorithms, to mitigate the O(N^2) bottleneck.
  - Investigate the use of learned or task-aware similarity metrics to better capture the functional relationships between experts, potentially leading to more effective merging strategies.
- **Broaden the experimental evaluation**
  - Evaluate PuzzleMoE on a more diverse set of MoE architectures, including models with different gating mechanisms (e.g., noisy top-k) or expert designs, to demonstrate the method's robustness and generalizability.
  - Extend the evaluation to a wider range of tasks, such as code generation (HumanEval), mathematical reasoning (GSM8K), or multilingual benchmarks, to showcase the method's applicability beyond the tested domains.

---

> ### Author Response · Authors · 2025-11-19
> **[1/2] Rebuttal**
>
> ## **Weakness1 and Question1: Limitations of the Learnable Unmerging Mechanism**
> We would like to humbly clarify that in PuzzleMoE, the unmerging step does not introduce any learnable parameters. **There are no trainable “unmerging coefficients”**. Instead, unmerging is a **deterministic, training-free reconstruction** implemented inside our custom CUDA kernel by decoding the bit-packed masks and sign bits. Consequently:
>
> (1). The parameter count of the compressed model is unchanged by unmerging.
>
> (2). There is no separate optimization step and therefore no additional hyperparameters associated with learning unmerging coefficients.
>
> The only extra cost comes from a few lightweight bit operations to decode the packed masks and signs for the activated experts. This decoding is fused into the GEMV kernel and performed on-the-fly only for routed experts, so the overall complexity remains dominated by the dense matrix multiplications. Our end-to-end latency measurements across different scales (demonstrated in Figure 4(c) in the manuscript) already include this decoding cost and still show clear speedups over the uncompressed MoE and other existing expert merging/dropping methods, indicating that the overhead is negligible in practice.
>
> ## **Weakness2 and Question2: Overhead of Computing Similarity Metrics and Grouping Experts**
>
> We thank the reviewer for this insightful comment and for highlighting the potential complexity concern. We would first like to clarify that PuzzleMoE **does not** require an all-pairs $O(N^{2})$ similarity computation. In our method, expert pairs are *randomly pre-assigned* for merging, and we do not perform any exhaustive clustering or LSH-style search over all experts. Figure 4(a) in the manuscript reports the end-to-end compression time, which already includes the cost of similarity computation and shows that the overall procedure is very fast in practice: Mixtral-8x7B with 8 experts can be compressed in about 2 minutes, and DeepSeek-MoE with 64 experts in about 10 minutes.
>
> Regarding the similarity metric itself, we do **not** use cosine similarity over weights. As defined in Eq. (1) of the manuscript, we adopt a per-entry magnitude-based similarity using the symmetric percent difference. This metric focuses on entries with comparable magnitudes while being insensitive to sign, thus avoiding penalizing weight pairs that differ only by sign and reducing distortion when reconstructing the original experts at inference time. It also involves only simple element-wise operations, making it very efficient to compute.
>
> Formally, let (N) be the number of experts and $D$ the number of parameters per expert. Since we compute similarity **only for the randomly assigned pairs**, there are $N/2$ pairs, and the cost per pair is $ O(D) $. The total complexity is $O\left(\frac{N}{2} \cdot D\right) \approx O(ND),$ which scales *linearly* with the number of experts and avoids the quadratic $ O(N^2) $ cost.
>
> To further substantiate this, we profiled the runtime of applying PuzzleMoE to Mixtral and DeepSeek-MoE with achieving 50% sparsity on A100 GPUs. As shown in the table below, the similarity computation step (e.g., 0.014 s $\times$ 32 for Mixtral and 0.017 s $\times$ 26 for DeepSeek-MoE) accounts for only a tiny fraction of the total compression time (121s and 585s, respectively). This confirms that expert grouping and similarity computation introduce negligible overhead in practice.
>
> | Model                     | Others | Similarity calculation time (per layer × #layers) | Merge time (per layer × #layers) | Forward time (per layer × #layers) | Total time |
> |---------------------------|--------|---------------------------------------------------|-----------------------------------|-------------------------------------|-----------|
> | Mixtral-8x7B-v0.1     | 6s     | 0.014s × 32                                      | 1.1s × 32                        | 2.5s × 32                          | 121s      |
> | DeepSeekMoE-16B | 16s    | 0.017s × 26                                      | 13.2s × 26                       | 8.7s × 26                          | 585s      |

---

> ### Author Response · Authors · 2025-11-19
> **[2/2] Rebuttal**
>
> ## **Weakness3 and Question3: Generalization of PuzzleMoE to Other Tasks and MoE Architectures Beyond Mixtral**
>
> Thank you for raising concerns about the breadth of our experimental evaluation. We would like to clarify that beyond Mixtral, **our paper already reports comprehensive results on DeepSeek-MoE, Qwen1.5-MoE, and Qwen3-MoE variants**, covering a range of scales and routing configurations. This provides strong evidence that PuzzleMoE generalizes across several popular MoE families. In fact, the set of architectures we evaluate is broader than those considered in prior MoE compression works such as NAEE, HC-SMoE, and D2-MoE.
>
> At present, mainstream deployed MoE LLMs predominantly use top-k gating. Architectures with substantially different gating mechanisms are still under active research and, to the best of our knowledge, are not yet widely adopted in production systems. Extending PuzzleMoE to these emerging architectures is an interesting direction that we leave for future work.
>
> We have also expanded our evaluation beyond standard language modeling and commonsense reasoning. Specifically, we now include challenging reasoning benchmarks (MATH500, AIME’24, AIME’25) and a code generation benchmark (HumanEval) as shown in the following two Tables. Across these tasks, PuzzleMoE consistently matches or outperforms the dense models and existing MoE compression baselines, achieving state-of-the-art performance among compressed MoE models. This demonstrates that PuzzleMoE remains effective not only on general language tasks but also on demanding mathematical reasoning and code generation workloads.
>
> | Model                          | HumanEval Pass@1 |
> |--------------------------------|------------------|
> | Qwen1.5-MoE-A2.7B (0% sparsity)              | 49.4             |
> | PuzzleMoE (25%)      | 45.1             |
> | HC-SMoE (25%)        | 11.0             |
> | PuzzleMoE (50%)      | 39.6             |
> | HC-SMoE (50%)        | 0                |
>
> | Model                      | MATH-500 | AIME24 | AIME25 |
> |----------------------------|----------|--------|--------|
> | Qwen3-MoE-30B-A3B (0% sparsity)                | 97.2     | 83.3   | 72.9   |
> | HC-SMoE (25%)      | 24.6     | 0.0      | 0.0      |
> | PuzzleMoE (25%)     | 96.2     | 71.1   | 61.5   |
>
> We have also updated this discussion and results in Section 4.3 in the submitted manuscript (in red text).

---

> > ### Comment · Reviewer_DoU5 · 2025-11-25
> >
> > Thank you address my concerns. I had raised my scores.

---

### Official Review · Reviewer_WnJg · 2025-11-01

**Soundness:** 3
**Presentation:** 3
**Contribution:** 2
**Rating:** 4
**Confidence:** 4

**Summary:**

The paper proposes PuzzleMoE, a training-free method to compress Mixture-of-Experts (MoE) models by addressing their large memory footprint. The method relies on two core innovations. First, it introduces a fine-grained, pairwise sparse expert merging algorithm. This algorithm uses a dual-mask system to create a merged expert: (1) a similarity-based mask identifies and averages shared, redundant weights between two experts, and (2) an activation-based saliency mask preserves critical, specialized weights from the more salient expert. Second, to avoid the significant storage overhead of these fine-grained masks and sign bits, it introduces a bit-packing scheme. This scheme exploits the narrow distribution of Bfloat16 exponent values, freeing 3 bits to store the 4 bits of required metadata (two masks, two signs) within the existing 16-bit weight representation. Inference is performed by a custom CUDA kernel that decodes this information on the fly.

**Strengths:**

* The method is training-free and exceptionally fast. Compressing Mixtral-8x7B takes only 2 minutes, drastically outperforming the 90 minutes required for search-based NAEE or 55 minutes for SVD-based D2.
* The dual-mask merging strategy is highly effective, demonstrating state-of-the-art accuracy. At 50% compression, it shows minimal degradation, while prior methods suffer catastrophic accuracy drops. On Mixtral-8x7B MMLU, PuzzleMoE achieves 65.7%, whereas NAEE and HC-SMOE collapse to 47.3% and 49.0%, respectively.
* The bit-packing of metadata into underutilized Bfloat16 exponent bits is a clever solution to the overhead problem that typically makes fine-grained sparsity impractical. This co-design enables practical, efficient inference, delivering up to 1.28x speedup on Mixtral.
* The method appears robust, showing low sensitivity to the choice of calibration dataset (C4 vs. Math) and the expert grouping strategy (random vs. searched), which simplifies its practical deployment.

**Weaknesses:**

* The bit-packing scheme is the method's primary strength but also its critical weakness. It is rigidly tied to 2-to-1 pairwise merging. As the ablation in Table 6 confirms, merging 3+ experts is infeasible because the required metadata (5+ bits) exceeds the 4 bits available (1 sign + 3 freed exponent). This fundamentally limits PuzzleMoE to a fixed 50% expert compression ratio (or 75% at 25% sparsity), lacking the flexibility of other pruning methods.
* The paper's claim of compatibility with quantization is not well-supported by its own data. In Table 9, the proposed PuzzleMoE (50%) + 3-bit Quant (resulting in 3.35 avg bits) achieves lower MMLU accuracy (72.4) on Mixtral-8x7B than the simpler 3-bit AWQ baseline alone (73.0 accuracy at 3.25 avg bits). This suggests that at this compression level, standard quantization is more effective, and the sparse merging adds complexity for a net performance loss.
* The comparison to Wanda (2:4) is inappropriate. PuzzleMoE implements fine-grained, unstructured sparsity that requires a custom kernel, whereas Wanda (2:4) is a structured sparsity pattern designed for native hardware acceleration. A 50% unstructured magnitude or Wanda pruning baseline is conspicuously missing. Without this comparison, it is impossible to know if the complex dual-mask merging algorithm is actually superior to simpler fine-grained pruning criteria when run with a similar custom kernel.
* The method relies on a saliency metric ($|W| \cdot ||X||_2$) directly borrowed from Wanda19. The ablation in Table 10 only compares this against an even simpler magnitude-only metric20. It fails to investigate any MoE-specific saliency signals, such as router gating scores, which might be more effective at identifying and preserving expert specialization.

**Questions:**

1) The proposed bit-packing scheme, which frees 3 bits by shifting the 8-bit exponent, is a key innovation. However, this seems to rigidly tie the method to a 2-to-1 merge (requiring 4 metadata bits, packed into the 1 sign + 3 freed bits). How can PuzzleMoE be adapted to other compression ratios (e.g., 3-to-1 for 67% sparsity) without abandoning this core bit-packing efficiency?
2) Table 9 indicates that for Mixtral-8x7B, 3-bit AWQ (a baseline) achieves 73.0 MMLU, while PuzzleMoE + 3-bit (the proposed method) achieves 72.4 MMLU. This suggests that at this low precision, the proposed sparse merging is actually detrimental. Could you clarify why this performance drop occurs and justify the combined approach?
3) The paper's baseline for 50% sparsity is 2:4 structured Wanda4. A more direct comparison would be 50% unstructured magnitude pruning (using the Wanda metric $|W| \cdot ||X||_2$) implemented with the same custom bit-decoding CUDA kernel. Was this baseline tested? It is essential for isolating the gains of the dual-mask merging algorithm from the gains of the custom kernel itself.
4) The saliency mask uses an activation-based metric from Wanda. Did the authors experiment with MoE-specific signals, such as the router's gating frequencies or weights, to define expert specialization? This seems like a more direct method for identifying critical, expert-specific parameters than a generic saliency score.

---

> ### Author Response · Authors · 2025-11-19
> **[1/2] Rebuttal**
>
> We greatly appreciate your positive feedback and constructive suggestions/questions for PuzzleMoE, and have addressed each of your comments in detail below:
>
> ## **Weakness1 and Question1: Limited flexibility of bit-packing scheme and how to extend PuzzleMoE to other compression ratios without losing bit-packing efficiency**
>
> Thank you very much for raising the concerns about the bit-packing scheme. We would like to clarify the roles of the two components in PuzzleMoE:
>
> 1. **Dual-mask expert merging algorithm** : This is the core algorithmic contribution that determines how experts are selected and merged.
>
> 2. **Bit-packing scheme that reuses bfloat16 sign/exponent bits** : This is a systems optimization to store merge masks more memory-efficiently at scale.
>
> The current bit-packing scheme is deliberately designed for 2-to-1 merging at 25% and 50% sparsity to **follow the prior MoE merging works such as NAEE, HC-SMoE, and SubMoE, which commonly study the 25% and 50% sparsity settings**. We therefore adopt the same settings for a fair comparison. Under these exact sparsity levels, **PuzzleMoE achieves much smaller accuracy degradation (≈2%), while prior methods often suffer very large drops (on average >20%) at 50% sparsity** as shown in Table 2 in the manuscript, making them difficult to deploy in practice.
>
> We acknowledge that the current bit-packing scheme is indeed restricted to 2-to-1 merging. However, this **does not limit PuzzleMoE’s ability to operate at more aggressive sparsity levels**. If we store the masks and signs explicitly without packing them into bfloat16, the same dual-mask merging algorithm can in principle support arbitrary compression ratios (e.g., 3-to-1, 4-to-1). In other words, the bit-packing scheme is introduced solely to make high-sparsity operation memory-efficient and practical, **not to constrain the algorithm or the achievable sparsity.**
>
> To further demonstrate both the effectiveness of PuzzleMoE at higher sparsity, we conduct additional experiments on DeepSeek-MoE where 4 experts are merged into 1 via hierarchical 2-to-1 merging, achieving a final sparsity of 75%. The corresponding results are reported in the table below. **PuzzleMoE consistently outperforms the existing SOTA HC-SMoE across nearly all benchmarks and improves the average score from 35.7 to 50.4 (+14.7 points).** Considering the superior performance of PuzzleMoE against existing expert merging works, designing a more flexible packed mask-storage scheme for sparsity levels beyond 50% is an interesting systems optimization that we leave for future work.
>
> | Method      | wiki   | arc-c | arc-e | hella | piqa | boolq | wino | mmlu | avg  |
> |------------|--------|-------|-------|-------|------|-------|------|------|------|
> | PuzzleMoE | **14.04** | **33.4** | **65.3** | **54.1** | **72.2** | **62.2** | **66.4** | **24.9** | **50.4** |
> | HC-SMoE   | 8261   | 19.9  | 27.1  | 26.3  | 54.3 | 59.8  | 50.8 | 23.6 | 35.7 |
>
> This has also been clarified in Appendix B.4 in the revised manuscript.
>
>
> ## **Weakness2 and Question2: Quantization compatibility and performance comparison between quantized PuzzleMoE and AWQ**
>
> Thank you very much for raising the concern about PuzzleMoE’s compatibility with quantization. We would like to clarify that the data in Table 9 (72.4 vs 73.0) shows the **average accuracy across 7 benchmarks**, not the MMLU accuracy.
>
> We acknowledge that although pure quantization (AWQ) performs slightly better on Mixtral-8×7B at very high compression, its performance lags behind PuzzleMoE on DeepSeek-MoE. This difference arises because the two techniques target different sources of redundancy: PuzzleMoE removes inter-expert redundancy through expert merging, whereas quantization primarily reduces intra-expert redundancy within each expert. **Therefore, we humbly clarify that PuzzleMoE is not designed to compete with quantization methods, but rather to complement them along an orthogonal compression dimension.**
>
> To further demonstrate the compatibility and effectiveness of PuzzleMoE with quantization compared to existing MoE merging algorithms, we report additional results where we combine NAEE and HC-SMoE with AWQ, matching the final compressed model to an average bit of 3.3 as shown in the Table below. For Mixtral-8×7B, QPuzzleMoE improves average accuracy **from 70.1 (NAEE+AWQ) to 72.4**. For DeepSeek-MoE, QPuzzleMoE outperforms HC-SMoE+AWQ, increasing the score **from 53.3 to 61.6**. These results demonstrate that QPuzzleMoE provides the best compression-accuracy trade-off than combining quantization with existing MoE merging/dropping methods.

---

> ### Author Response · Authors · 2025-11-19
> **[2/2] Rebuttal**
>
> | Model       | Method                                | wiki  | arc-c | arc-e | hella | piqa | boolq | wino | mmlu | avg   |
> |------------|----------------------------------------|-------|-------|-------|-------|------|-------|------|------|-------|
> | Mixtral-8x7B-v0.1     | PuzzleMoE-50% sparse + 3-bit quant     | **4.5**  | **53.3** | **82.4** | **63.2** | **81.6** | **85.5** | 75.9 | **65.2** | **72.4** |
> | Mixtral-8x7B-v0.1    | NAEE-25% sparse + 4-bit quant          | 5.1   | 51    | 81    | 61.4  | 81.2 | 82.8  | 75.9 | 58   | 70.1  |
> | Mixtral-8x7B-v0.1    | HCSMoE-25% sparse + 4-bit quant        | 5.39  | 50.6  | 78.8  | 61.1  | 80.5 | 84.5  | **76**   | 59.2 | 70.1  |
> | DeepSeekMoE-16B| PuzzleMoE-50% sparse + 3-bit quant     | **7.05** | **42.3**  | **74.8**  | **55.8**  | **78.4** | **74.6**  | **70.2** | **35.2** | **61.6**  |
> | DeepSeekMoE-16B| HCSMoE-25% sparse + 4-bit quant        | 26.7  | 35.3  | 64.1  | 43.6  | 72.9 | 66.3  | 65.8 | 25   | 53.3  |
>
> We have also updated this discussion and results in Appendix B.3.1 in the submitted manuscript (in red text).
>
> ## **Weakness3 and Question3: Need 50% unstructured pruning baseline to fairly compare PuzzleMoE against Wanda**
>
> Thank you very much for pointing out this important comparison. We agree that 50% unstructured pruning using the Wanda metric is a crucial baseline for validating the dual-mask merging mechanism, and the corresponding results are reported in the table below. However, as discussed in https://openreview.net/forum?id=iso0KV2HVq, **there is currently no efficient CUDA kernel support for 50% unstructured sparsity on GPUs**. Consequently, such unstructured sparsity does not translate into actual memory savings or inference speedup in practice. By contrast, PuzzleMoE not only achieves better accuracy than 50% unstructured Wanda pruning, but also yields practical speedups on GPU thanks to its structured expert-level sparsity.
>
> | Model       | Method                         | wiki  | arc-c | arc-e | hella | piqa | boolq | wino | mmlu | avg   |
> |------------|---------------------------------|-------|-------|-------|-------|------|-------|------|------|-------|
> | Mixtral-8x7B-v0.1    | PuzzleMoE-50%                  | **4.36** | **53.8** | **82.4** | **63.3**  | **81.7** | 85.3 | 75.8 | 65.7 | **72.6** |
> | Mixtral-8x7B-v0.1    | Unstructured-50%               | 4.68  | 53.1  | 81.8  | 62.4  | 81.4 | **85.9** | **76.6** | **65.8** | 72.4  |
> | DeepSeekMoE-16B| PuzzleMoE-50%                  | **6.88** | **43.0**    | **75.2**  | **56.3**  | **78.4** | 74.5 | 70.3 | **36.9** | **62.1**  |
> | DeepSeekMoE-16B| Unstructured-50%               | 7.18  | 41.4  | 74.8  | 54.8  | 77.9 | **76.3** | **70.6** | 35.4 | 61.6  |
>
> ## **Weakness4 and Question4: Reliance on Wanda saliency metric instead of exploring MoE-specific signals**
>
> To address your concern, we additionally evaluated several MoE-aware saliency variants that incorporate router information to different extents: W (\$|W|$), WG (\$|W| \cdot ||gate||_2$), WA (\$|W| \cdot ||X||_2$, same as the original Wanda metric), and WAG (\$|W| \cdot ||X \cdot gate||_2$).
>
> As shown in the table below, the activation-based variants (WA and WAG) achieve very similar overall performance and consistently match or slightly outperform the weight-only variants (W and WG). In contrast, adding gating scores on top of weight-only saliency (WG) brings almost no benefit. **This indicates that the main performance gains come from incorporating activation information into the saliency metric, which better preserves expert specialization, while additional router-gating signals provide no significant improvements.**
>
> (W – \$|W|$, WG – \$|W| \cdot ||gate||_2$, WA – \$|W| \cdot ||X||_2$, WAG – \$|W| \cdot ||X \cdot gate||_2$)
>
> | Model       | Method              | wiki | arc-c | arc-e | hella | piqa | boolq | wino | mmlu | avg  |
> |------------|---------------------|------|-------|-------|-------|------|-------|------|------|------|
> | Mixtral-8x7B-v0.1    | PuzzleMoE-50%-W     | 4.38 | 53.6  | 82.5  | 63.4  | 81.7 | 85.3  | 75.8 | 65.8 | 72.6 |
> | Mixtral-8x7B-v0.1    | PuzzleMoE-50%-WG    | 4.38 | 53.6  | 82.4  | 63.3  | 81.7 | 85.3  | 75.8 | 65.8 | 72.6 |
> | Mixtral-8x7B-v0.1    | PuzzleMoE-50%-WA    | 4.36 | 53.8  | 82.4  | 63.3  | 81.7 | 85.3  | 75.8 | 65.7 | 72.6 |
> | Mixtral-8x7B-v0.1    | PuzzleMoE-50%-WAG   | 4.36 | 53.9  | 82.3  | 63.3  | 81.7 | 85.4  | 75.7 | 65.7 | 72.6 |
> | DeepSeekMoE-16B| PuzzleMoE-50%-W     | 6.98 | 42.4  | 75.2  | 56.0  | 78.4 | 75.1  | 70.1 | 35.4 | 61.8 |
> | DeepSeekMoE-16B| PuzzleMoE-50%-WG    | 6.96 | 42.1  | 75.2  | 56.1  | 78.2 | 75.4  | 70.1 | 35.1 | 61.7 |
> | DeepSeekMoE-16B| PuzzleMoE-50%-WA    | 6.88 | 43.0  | 75.2  | 56.3  | 78.4 | 74.5  | 70.3 | 36.9 | 62.1 |
> | DeepSeekMoE-16B| PuzzleMoE-50%-WAG   | 6.89 | 43.1  | 75.2  | 56.1  | 78.5 | 74.8  | 70.3 | 36.8 | 62.1 |
>
> We have also updated this discussion and results in Appendix B.3.2 in the submitted manuscript (in red text).

---

### Author Response · Authors · 2025-12-02
**[1/2] Summary for AC after Score Rollback**

Dear Area Chair,

Given the unique circumstances of the review process, we would like to summarize the paper, prior reviews, and what we added in the rebuttal so that you have a concise view of the current status.

---
>**Strengths of our submission shared by reviewers:**
---

**1. Strong Novelty.**

We propose **the first fine-grained, element-wise pairwise merging approach** for MoE LLMs expert merging, which is a novel contribution relative to existing coarse merging or dropping strategies for MoE LLM expert compression.

**2. State-of-the-art performance.**

PuzzleMoE delivers **superior accuracy** under substantial compression for MoE LLMs. Particularly, at 50% sparsity, PuzzleMoE maintains strong performance while prior methods degrade drastically, demonstrating robust preservation of expert knowledge and minimal task-level accuracy loss.

**3. Innovative systems co-design.**

Bit-packing metadata into underutilized Bfloat16 exponent bits **effectively eliminates sparsity overhead** introduced by the fine-grained merging process and yields up to 1.28× inference speedup. This systems-aware design overcomes a key obstacle in fine-grained MoE compression and enables efficient real-world inference.

**4. Training-free, extremely efficient compression.**

PuzzleMoE is **training-free and achieves exceptionally fast compression process**. It can compress Mixtral-8×7B in only two minutes, far outperforming previous works involving search-based or SVD-based baselines. This efficiency makes PuzzleMoE highly practical for real-world deployment.

---
>**We summarize the major concerns proposed by reviewers and how we address them in the rebuttal:**
---
**Concern1: PuzzleMoE’s flexibility on more aggressive sparsity. (WnJg, CUvt)**

+ We justified that although the bit-packing method supports only 50% and 25% expert sparsity, the underlying dual-mask merging algorithm can theoretically support higher sparsity if the masks are stored explicitly.

+ We shown that PuzzleMoE consistently outperforms the existing SOTA method across nearly all benchmarks and improves the average score from 35.7 to 50.4 with 75% expert sparsity.

+ We discussed that expanding the bit-packing mechanism to allow for more aggressive sparsity is an interesting area for future work.

**Concern2: PuzzleMoE’s compatibility with quantization methods. (WnJg)**

+ We justified that PuzzleMoE does not directly compete with quantization techniques, but rather **complements them** by targeting inter-expert redundancy through expert merging.

+ We shown that PuzzleMoE method **outperforms other MoE-specific compression methods** (e.g. NAEE, HC-SMoE) when combined with quantization, and can achieve on-par pure quantization methods like AWQ at very high compression (e.g., 3-bit quantization). Thus, the combination of PuzzleMoE and quantization should be viewed as **orthogonal approaches** that work together for better compression.


**Concern3: Generalization on different models and tasks. (DoU5)**

+ In the original submission, PuzzleMoE has already been evaluated on a wide range of models, including Mixtral, Deepseek-MoE, Qwen1.5-MoE, and Qwen3-MoE. PuzzleMoE has also demonstrated strong performance in language modeling (Wikitext2) and commonsense reasoning (MMLU, ARC, PIQA, etc.).

+ In the rebuttal, we further evaluated PuzzleMoE on coding tasks (HumanEval) and mathematical reasoning tasks (GSM8K, MATH500, AIME), demonstrating PuzzleMoE's robustness and generalization capabilities compared to previous works.

**Concern4: Experimental settings and practical speedup. (SRNA, CUvt)**

+ We clarified that the primary goal of all MoE compression methods is to **reduce the memory requirement of MoE inference**, the number of activated experts remains the same while the total expert parameters are reduced.

+ PuzzleMoE's speedup is primarily achieved when **transitioning from multi-GPU to single-GPU deployments**. This speedup is driven by the reduction in total expert parameters, **enabling deployment on fewer GPUs and minimizing expensive cross-GPU communications**. The reported latency reduction (~1.2×) is significant in multi-GPU setups where previous models required multiple GPUs for inference.

---

> ### Author Response · Authors · 2025-12-02
> **[2/2] Summary for AC after Score Rollback**
>
> ---
> >**Summary**
> ---
> Even though the initial scores for our submission are borderline, we believe our rebuttal has thoroughly addressed the reviewers’ concerns. We would also like to emphasize that, as the first fine-grained, element-wise pairwise expert-merging method, PuzzleMoE naturally has some limitations, just as other approaches do. However, **its state-of-the-art accuracy, the novel bit-packing mechanism, and its strong compression efficiency compared to existing methods together open a promising direction for MoE LLM expert merging.** Therefore, we respectfully hope you will consider our work carefully in your final decision.
>
> We would of course be happy to answer any further questions or provide additional clarifications if that would be helpful for your evaluation. Thank you very much for your time and consideration.
>
> Best regards,
>
> Authors of submission 4218

---

### Meta-Review · Area_Chair_HV6F · 2025-12-19

**Summary:**

The submission proposes PuzzleMoE, a training-free framework for compressing Mixture-of-Experts (MoE) models by merging pairs of experts. The method employs a dual-mask strategy to retain salient and similar weights between expert pairs and introduces a system-level optimization that packs the resulting mask and sign bits into the unused exponent bits of BFloat16 tensors. Experiments on Mixtral, DeepSeek-MoE, and Qwen-MoE models demonstrate that the method can reduce memory usage by approximately 50% with relatively minor accuracy degradation compared to some existing merging baselines, while offering a modest inference speedup via a custom CUDA kernel.

**Reviewer Concerns:**

Reviewers raised several valid concerns that persist despite the rebuttal. A primary technical limitation identified is the rigidity of the bit-packing scheme, which restricts the method to a fixed 50% compression ratio (2-to-1 merging), lacking the flexibility of standard quantization or unstructured pruning. Furthermore, the empirical advantage over standard quantization methods (e.g., AWQ) was questioned; reviewers noted that simple quantization often achieves comparable or better performance-efficiency trade-offs without the implementation complexity of sparse merging. While the authors argued that their method is orthogonal to quantization, the added complexity of the dual-mask pipeline was viewed as a hurdle. Additionally, there were initial concerns regarding the "unmerging" overhead and the scalability of pairwise merging, though the authors clarified the deterministic nature of their kernel.


From the Area Chair's perspective, there are significant reservations regarding the core motivation and architectural assumptions of the work. Prior literature establishes that in expert merging, the grouping process is typically the computationally expensive bottleneck, whereas the actual merging operation  is comparatively efficient. The authors circumvent this bottleneck by employing random grouping, yet this design choice raises fundamental doubts about efficacy. Specifically, there are strong reservations about whether sparsity-based merging methods remain effective when integrating experts that exhibit low similarity. In MoE models, experts are theoretically specialized; if experts are dissimilar (as is often the case with random pairing), forcing them to merge via sparsity is likely to degrade the unique semantic capabilities of the model. The paper does not sufficiently address how random grouping avoids the destructive interference of merging highly distinct experts, which is a critical oversight in the method's motivation.

**Reviewer Scores:**

Reviewer WnJg (Score: 4) maintained their score, as the concern regarding the method's rigidity and the unfavorable comparison to simple AWQ baselines remains a significant drawback.

Reviewer DoU5 (Score: raised to Weak Accept during discussion) improved their score after the authors clarified the overhead of the unmerging mechanism, but likely remains cautious about the overall impact.

Reviewer SRNA (Score: 4) maintained their assessment, viewing the work as an incremental combination of existing techniques with limited inference acceleration.

Reviewer CUvt (Score: 4) maintained their score, as the concern regarding the theoretical floor of 50% compression and the lack of flexibility in the bit-packing scheme was not fundamentally resolved.

---

### Decision · Program_Chairs · 2026-01-26

Reject